# TGFβ signaling curbs cell fusion and muscle regeneration

Francesco Girardi[1], Anissa Taleb[1], Majid Ebrahimi[2,3], Asiman Datye[2,3], Dilani G. Gamage[4], Cécile Peccate[1], Lorenzo Giordani[1], Douglas P. Millay [4,5], Penney M. Gilbert[2,3,6], Bruno Cadot [1] & Fabien Le Grand [1,7✉]

Muscle cell fusion is a multistep process involving cell migration, adhesion, membrane remodeling and actin-nucleation pathways to generate multinucleated myotubes. However, molecular brakes restraining cell–cell fusion events have remained elusive. Here we show that transforming growth factor beta (TGFβ) pathway is active in adult muscle cells throughout fusion. We find TGFβ signaling reduces cell fusion, regardless of the cells' ability to move and establish cell-cell contacts. In contrast, inhibition of TGFβ signaling enhances cell fusion and promotes branching between myotubes in mouse and human. Exogenous addition of TGFβ protein in vivo during muscle regeneration results in a loss of muscle function while inhibition of TGFβR2 induces the formation of giant myofibers. Transcriptome analyses and functional assays reveal that TGFβ controls the expression of actin-related genes to reduce cell spreading. TGFβ signaling is therefore requisite to limit mammalian myoblast fusion, determining myonuclei numbers and myofiber size.

[1] Sorbonne Université, INSERM UMRS974, Association Institut de Myologie, Centre de Recherche en Myologie, 75013 Paris, France. [2] Institute of Biomedical Engineering, University of Toronto, Toronto, ON M5S3G9, Canada. [3] Donnelly Centre for Cellular and Biomolecular Research, Toronto, ON M5S3E1, Canada. [4] Division of Molecular Cardiovascular Biology, Cincinnati Children's Hospital Medical Center, Cincinnati, OH 45229, USA. [5] Department of Pediatrics, University of Cincinnati College of Medicine, Cincinnati, OH 45229, USA. [6] Department of Cell and Systems Biology, University of Toronto, Toronto, ON M5S3G5, Canada. [7] Present address: Institut NeuroMyoGène, Université Claude Bernard Lyon 1, CNRS UMR 5310, INSERM U1217, 69008 Lyon, France. ✉email: fabien.le-grand@inserm.fr

The adult skeletal muscle cell is a syncytial myofiber that contains hundreds of myonuclei. Formation and regeneration of the myofiber requires fusion of mononuclear progenitors (myoblasts) to form multinucleated myotubes. Located in a niche around the myofibers are quiescent muscle stem cells[1], called satellite cells, which can activate and proliferate to give rise to adult myoblasts. Myoblasts differentiate into myocytes competent to fuse with each other and with myofibers[2]. As such, cell fusion plays essential roles in the adult, allowing physiological muscle hypertrophy[3,4] and muscle regeneration following injury[5,6].

When induced to fuse, adult myoblasts exit the cell cycle, commit to terminal differentiation, and migrate toward each other[7]. They then adhere through membrane integrins[8] and cadherins[9]. The later stages of fusion are controlled by the muscle-specific protein Myomaker[10] and peptide Myomerger[11–13] (also known as MINION, Myomixer). Myomaker and Myomerger function in hemifusion and pore formation activity, respectively. Recent studies demonstrated that muscle cell fusion is promoted by actin-based structures[14] generating protrusive forces[15] and membrane stress before coalescence[16]. The fusion process must be tightly controlled to ensure that fusogenic myoblasts do not form aberrant hypertrophic syncytia or fuse with non-muscle cells. However, while it is known that muscle cell fusion can be prevented by tetraspanins at the cell membrane[17], few signaling pathways that can limit this process and prevent unscheduled cell fusion has been identified.

Canonical Wnt/β-catenin signaling is a crucial regulator of satellite cells and adult muscle regeneration[18,19]. Interestingly, β-catenin activation in adult myoblasts promotes their progression through the myogenic lineage[20], but limits differentiated muscle cell fusion[6]. This contrasting regulation of different steps of adult myogenesis can be explained by the fact that β-catenin signaling induces the expression of transforming growth factor beta (TGFβ) ligands and receptors by adult muscle cells[20,21]. As such, a large body of evidence, built on seminal work from three decades ago, clearly demonstrated that TGFβ signaling inhibits muscle cell terminal differentiation[22,23]. Mechanistically, TGFβ signaling has been shown, mainly in vitro, to negatively regulate differentiation through functional repression of the myogenic regulatory factors Myod1[24] and Myogenin[25]. However, TGFβ signaling has a broader function in muscle cells, including quiescence[26] and activation[27].

The impact of TGFβ signaling specifically in muscle cell fusion has not been investigated. This gap in our knowledge mainly comes from the fact that the primary effects of TGFβ over-expression in skeletal muscles are the development of endomysial fibrosis, due to its role as a potent growth factor for connective tissue cells[28,29].

We thus asked whether TGFβ signaling regulates the fusion of differentiated muscle cells. Here, we present a detailed analysis of its role during muscle cell differentiation and tissue regeneration. We observe that the activation of TGFβ signaling leads to a strong reduction of myotube formation, independently of previous differentiating steps such as myoblast motility and Myogenin expression. Importantly, neither myoblast proliferation nor mortality is altered by the signaling modulation. Inhibition of TGFβ signaling results in the formation of large syncytia with an elevated number of nuclei and aberrant shapes. These results are confirmed in vivo, where pharmacological stimulation of the signaling pathway dramatically reduces fiber size and myonuclear number, while its inhibition generates giant myofibers with multiple myonuclei. Collectively, our results demonstrate that TGFβ signaling controls syncytia formation and muscle regeneration by limiting muscle cell fusion.

## Results

**TGFβ signaling is active in adult muscle progenitor cells.** TGFβ ligands bind to TGFBR2 that will recruit a type I receptor dimer. The receptor complex will then phosphorylate SMAD2 and SMAD3 that will accumulate in the nucleus where they act as transcription factors[30]. To evaluate TGFβ isoforms expression by adult muscle progenitor cells, we purified limb muscle satellite cells and grew them in vitro as primary myoblasts. We observed that Tgfb1 and Tgfb2 expression levels were high in proliferating cells, and diminished following induction of differentiation, while Tgfb3 expression pattern showed an opposite trend (Fig. 1a), suggesting that TGFβ signaling may be still active at later differentiation time points. Of note, the expression levels of the TGFβ receptors (Alk5, also known as Tgfbr1; and Tgfbr2) did not significantly change during the course of in vitro myogenesis (Fig. 1b). We next investigated the state of TGFβ signaling in primary myoblasts, differentiated myocytes, and multinucleated myotubes. We observed that the expression level of the TGFβ/ SMAD2/3 target gene Smad7 diminished during myogenic progression (Fig. 1c). While immunolocalization of phosphorylated-SMAD2/3 (p-SMAD2/3) proteins showed that the canonical TGFβ pathway is active at all studied stages (Fig. 1d), quantitative western blotting experiments demonstrated that the intensity of TGFβ signaling decreases during muscle cell differentiation, but is not abrogated in multinucleated cells (Fig. 1e). Previous work has established that TGFβ ligands are secreted during muscle tissue repair[31]. Gene expression analysis of regenerating tibialis anterior (TA) muscles demonstrated that the three TGFβ isoforms are dynamically expressed following injury (Supplementary Fig. 1a). Evaluation of protein levels further validated that TGFβ1 and TGFβ3 expressions peak during the acute phase of tissue repair ~4 days post injury (d.p.i.), while TGFβ2 protein levels are higher at later time points (7 d.p.i.) (Supplementary Fig. 1b).

The dynamic expression profiles of TGFβ isoforms during adult myogenesis both in vitro and in vivo prompted us to evaluate the activity of TGFβ signaling in vivo at the cellular level. We performed co-immunolocalization for p-SMAD3 proteins and markers of muscle cells on cryosections of uninjured and regenerating TA muscles. In the resting muscle, p-SMAD3 proteins are detected in quiescent satellite cells and myonuclei (Fig. 2a). In acutely regenerating tissues, we found it to be active in proliferating myogenic stem and progenitor cells expressing PAX7 and in centrally located myonuclei of newly formed myofibers (Fig. 2b). In contrast, TGFβ signaling is inactive (p-SMAD3 staining at background levels) in MYOGENIN+ differentiating myocytes (Fig. 2d). At 7 d.p.i., when muscle architecture is restored, p-SMAD3 expression is detected in the nuclei of satellite cells and myonuclei, as well as in interstitial cells (Fig. 2c). While the presence of an active TGFβ signaling in PAX7-expressing cells was expected, the p-SMAD3 immunoreactivity of recently fused myonuclei was both surprising and fitting with the in vitro data. Together, these observations suggest that TGFβ pathway is involved in regulating the assembly of multinucleated muscle cells. We thus aimed to investigate the role of TGFβ in the fusion process.

**Fusion defects in adult muscle cells stimulated by TGFβ.** Since all three TGFβ ligands are expressed during adult myogenesis, we evaluated the impact of recombinant protein treatments on primary myoblast differentiation and fusion (Supplementary Fig. 2a). After 72 h of differentiation, muscle cells aggregated to form multinucleated myotubes, while the addition of recombinant TGFβ proteins forced muscle cells to remain mostly mononucleated (Supplementary Fig. 2b, e). Quantification of Myh3 gene expression, which codes for the embryonic myosin heavy-chain isoform, further indicated that the cells in TGFβ-treated cultures were in a less mature state than control cultures (Supplementary Fig. 2c). However, quantification of the

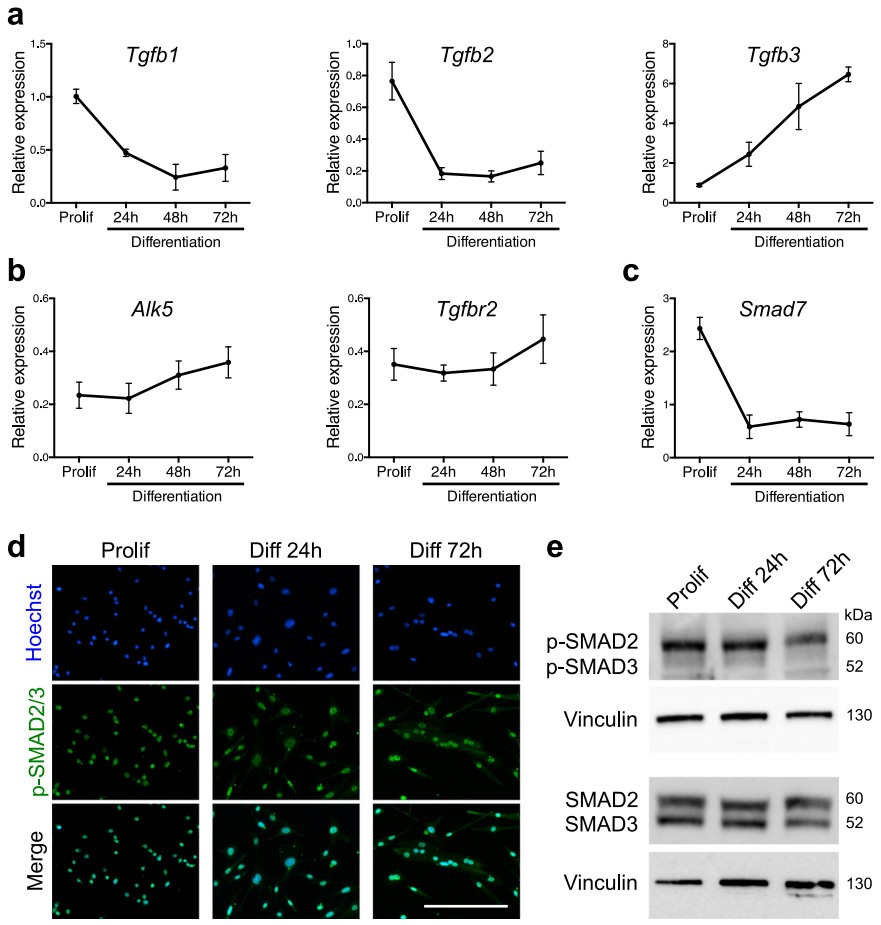

**Fig. 1 TGFβ signaling pathway remains active during myoblast differentiation. a** qRT-PCR analysis of *Tgfb1, 2,* and *3* transcripts expression during in vitro differentiation of primary muscle cells shows different profiles. $N = 3$ biologically independent experiments for each time point. **b** qRT-PCR analysis of *Alk5* and *Tgfbr2* transcript expressions describes a constant expression of the receptors during primary muscle cell differentiation. $N = 6$ biologically independent experiments for each time point. **c** qRT-PCR analysis of the TGFβ target gene *Smad7* transcript expression reveals a decreased activity of the pathway alongside in vitro primary muscle cell differentiation. $N = 3$ biologically independent experiments for each time point. **d** p-SMAD2/3 immunofluorescent staining of proliferating, differentiating, and differentiated primary myoblasts reveals a constant and basal activation of the pathway. $N = 3$ primary cell cultures. **e** p-SMAD2/3 and SMAD2/3 western-blot analysis of proliferating, differentiating, and differentiated primary myoblasts confirms a decrease in SMAD2/3 phosphorylation during differentiation. $N = 3$ biologically independent experiments. Scale bars: **d** 200 µm. Data are presented as mean ± SEM. Source data are provided as a Source Data file.

percentage of differentiated nuclei expressing Pan-MYOSIN HEAVY-CHAIN (Pan-MyHC) proteins revealed that the vast majority (>90%) of myoblasts did undergo differentiation in all conditions (Supplementary Fig. 2d), suggesting that TGFβ signaling does not primarily block muscle cell differentiation. Likewise, quantification of Pan-MyHC staining intensity of single multinucleated muscle cells demonstrated that TGFβ-treated myotubes, while containing less nuclei, expressed similar levels of the differentiation marker (Supplementary Fig. 3a).

To test this hypothesis, we adapted the protocol used by Saclier et al.[32] to uncouple differentiation and fusion of primary muscle cells. In this experimental setup, primary myoblasts were differentiated for 2 days at a low density that does not allow contact between cells. The cells were then split and re-plated at a high density and cultured for an additional 2 days to evaluate muscle cell fusion (Fig. 3a). Following re-plating, almost all muscle cells were terminally differentiated and expressed MYOGENIN (>94%) (Fig. 3b). To assert that TGFβ signaling does not impact the differentiation state of re-plated cells, we stimulated them with recombinant TGFβ1 protein and validated that the treatment did not result in changes in *Myogenin* gene expression, but did activate expression of the TGFβ target gene

*Smad7* (Fig. 3c). We thus evaluated the effect of TGFβ protein stimulations on mononucleated differentiated muscle cells (myocytes) and observed that all three TGFβ isoforms strongly inhibited cell fusion (Fig. 3d), despite the muscle cells progressing down the differentiation pathway (Fig. 3e; ~100% MyHC+). Activation of the TGFβ pathway reduced the fusion index (Fig. 3f) and completely blocked the formation of large myotubes (Fig. 3g), thus demonstrating that TGFβ signaling limits muscle cell fusion independently of myogenic differentiation. Moreover, quantification of Pan-MyHC staining intensity of single myotubes did not reveal any differences in the maturation states between control and TGFβ-stimulated cells (Supplementary Fig. 3b). Importantly, the addition of TGFβ proteins to adult muscle cells did not alter myoblast proliferation (Supplementary Fig. S4a), did not induce programmed cell death in myoblasts and myocytes (Supplementary Fig. 4b), and did not modify myoblasts and myocytes ability to migrate in scratch-wound assays (Supplementary Fig. 4c).

**TGFβ signaling does not impact cell–cell contacts.** To fuse, myocytes must undergo cell–cell recognition and adhesion[33,34].

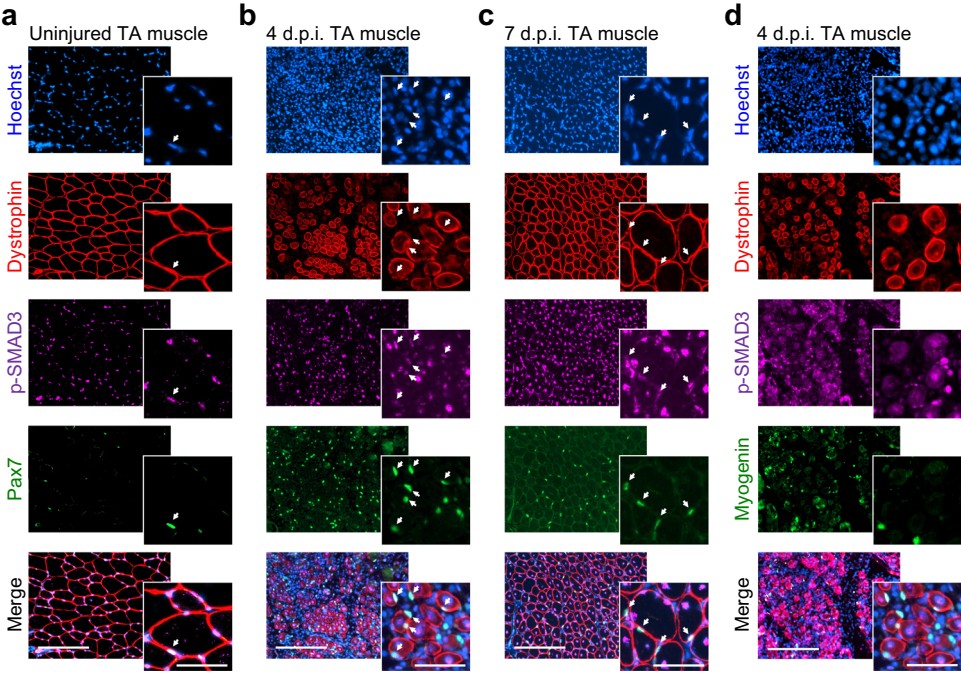

**Fig. 2 The state of TGFβ signaling during in vivo muscle regeneration. a–c** p-SMAD3, PAX7, and DYSTROPHIN immunofluorescent stainings on 0-, 4-, and 7-day post injury (d.p.i.) regenerating TA muscle cryosections. SMAD3 signaling is active in interstitial cells and PAX7+ cells (white arrows) during muscle regeneration. p-SMAD3 is strongly expressed by myonuclei of the regenerating myofibers marked by DYSTROPHIN at 4 d.p.i. $N = 8$ cryosections. **d** p-SMAD3, MYOGENIN, and DYSTROPHIN immunofluorescent staining on 4 d.p.i. regenerating TA muscle cryosections. Myocytes do not express p-SMAD3 at the time of differentiation. $N = 8$ cryosections. Scale bars: low magnification, 400 μm; high magnification, 100 μm.

Cell motility is therefore required to bring pre-fusing cells into proximity. We aimed to evaluate if TGFβ signaling impacts muscle cell motility and contact using live imaging. To this aim, we did not use the replating technique since high cell density would have impeded a clear tracking of cells' motions and contacts. We stimulated primary muscle cells with a single pulse of recombinant TGFβ1 protein (Fig. 4a) and validated that this treatment does not significantly change the proportion of MYOD1+ nuclei at 48 h in the differentiation medium (Fig. 4b) and the expression of Myogenin at 60 h in the differentiation medium (Fig. 4c). These observations suggest that the inhibitory action of TGFβ signaling on myoblast differentiation is partially compensated in culture, at least for the analyzed markers. We then recorded primary muscle cells at two different time frames (early differentiation: before cell fusion; late differentiation: during syncytium formation) to determine the effect of exogenous TGFβ1 protein (Fig. 4d) (Supplementary Movies 1–4). During early differentiation, TGFβ1 stimulation did not impact cell motility and did not change the frequency of cell–cell contacts (Fig. 4e and Supplementary Movies 1 and 2). Interestingly, during the late differentiation, TGFβ1-treated cells could still contact each other with a frequency similar to control cells, but these contacts were infructuous (Supplementary Movies 3 and 4). As such, we observed that TGFβ signaling prevented contacting cells from fusing and strongly increased the proportion of paired cells that remained mononucleated (Fig. 4e). Together, our observations indicate that TGFβ signaling does not impact "pre-fusion" cell dynamics, but prevents the merging of cells into syncytia.

**Inhibition of TGFβ receptor function promotes muscle cell fusion**. We next asked if inhibition of TGFβ signaling in fusing myocytes could enhance the formation of multinucleated myotubes. To this aim, we selected ITD-1, a highly selective TGFβ inhibitor that triggers proteasomal degradation of TGFBR2[35]. ITD-1 clears TGFBR2 from the cell surface and selectively inhibits intracellular signaling. ITD-1 treatment of primary myocytes resulted in reduced expression of TGFβ target genes (Fig. 5a) and blocked the phosphorylation of nuclear SMAD2/3 proteins induced by TGFβ1 treatment (Fig. 5b). Importantly, treatment of differentiated mononucleated muscle cells re-plated at high density (as in Fig. 3a) with ITD-1 enhanced the fusion process (Fig. 5c). As such, ITD-1-treated cultures showed a higher fusion index (Fig. 5e) and were composed of myotubes of larger diameter (Fig. 5f), containing more nuclei (Fig. 5d) and characterized by an aberrant branched shape (Fig. 5c, g). Since ITD-1 does not fully inhibit p-SMAD2/3 signaling in our assays, we further investigated SB-431542, another TGFβ inhibitor that targets ALK5/TGFBR1[36]. Treatment of re-plated myocytes with SB-431542 strongly increased the percentage of large myotubes containing high numbers of nuclei, suggesting that both type 1 and type 2 receptors are mediating the effects of TGFβ on muscle cell fusion (Supplementary Fig. 5). Taken together, our data demonstrate that TGFβ receptors regulate fusion and suggest that the levels of TGFβ signaling must be tightly controlled to ensure proper syncytia formation.

To test if TGFβ signaling controls fusion cell autonomously, we expanded primary myoblasts from satellite cells expressing either H2B-GFP or membrane tdTomato (pseudocolored in blue). Both primary cell types were pre-differentiated at low density, but only the GFP-expressing myocytes were treated with either TGFβ1 or ITD-1 and their F-actin content stained with SiR-Actin (pseudocoloured in red) (Fig. 6a). Cells were then mixed, re-plated at high density, and fusion events were imaged live (Fig. 6b). We observed that fusion of GFP-labeled myonuclei into tdTomato myotubes was controlled by the intrinsic state of TGFβ signaling in the fusing cells (Fig. 6c). As such the incidence of heterologous fusion was controlled by TGFβ signaling (Fig. 6d). Interestingly, we noticed that multinucleated myotubes could fuse together and that TGFβ signaling regulates the frequency of myotube-to-myotube fusion (Fig. 6e)

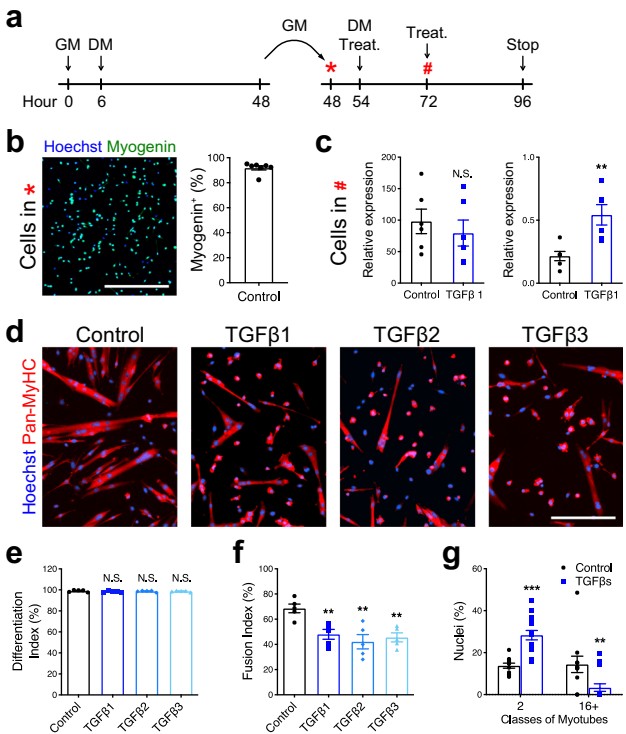

**Fig. 3 TGFβ signaling limits cell fusion. a** Experimental scheme. Primary myoblasts seeded at low density (5000 cells/cm²) were differentiated for 2 days, split, and re-plated at high density (75,000 cells/cm²) and cultured for 2 more days. **b** Immunofluorescent staining for MYOGENIN of primary myocytes pre-differentiated for 48 h and re-plated at high density confirms that >90% of cells express Myogenin. $N = 7$ biologically independent experiments. **c** qRT-PCR analysis for *Myogenin* and *Smad7* transcript expression of re-plated primary myocytes cultured for 24 h with or without TGFβ1 recombinant protein. Although TGFβ1 stimulation activates *Smad7* expression, it does not affect *Myogenin* transcript levels. $N = 6$ primary cell cultures. **d** Immunofluorescent staining for the MYOSIN HEAVY-CHAIN isoforms (Pan-MyHC) of re-plated primary myocytes cultured for 48 h. **e** Percentage of Pan-MyHC-expressing cells of re-plated myotubes shows that cells were differentiated in all conditions. $N = 5$ biologically independent experiments. **f** Fusion index of re-plated myotubes reveals that TGFβ stimulation inhibits fusion. $N = 5$ biologically independent experiments. **g** Percentage of nuclei in the smallest and largest myotube classes. TGFβ-treated myotubes are characterized by less nuclei per myotube. $N = 11$ (control) and 15 (TGFβ1) biologically independent experiments. Scale bars: **b** 400 µm; **d** 200 µm. Data are presented as mean ± SEM. Unpaired two-tailed Student's *t* tests were used to compare between data. ** and *** denote a significant difference with the Control group of $P < 0.01$ and $P < 0.001$, respectively. NS not significant. Source data are provided as a Source Data file.

(Supplementary Movies 5–7). These results suggest that TGFβ acts cell autonomously to limit the fusion between muscle cells, and to prevent fusion between syncytia.

**TGFβ signaling regulates the formation of human muscle microtissues (hMMTs).** To determine the human relevance of the hyper-fusion phenotype we observed with mouse cells, we evaluated the influence of TGFβ pathway inhibition on human muscle cell differentiation in a 3D culture format[37] (Fig. 7a). In this case, immortalized human muscle progenitor cells were embedded within a fibrin scaffold containing reconstituted basement membrane, which was spread within a custom polydimethylsulfoxane rubber device in which at either end of a cell seeding depression is

a vertical post (also cast using flexible rubber). Within hours of plating the cell-scaffold slurry within the depression, the cells self-organize across the axis of uni-axial tension supported by the vertical posts to generate a compact 3D tissue. By quantifying the tissue remodeling and compaction process (Fig. 7b, c), we found that inhibition of TGFBR1 function by the small-molecule SB-431542 did not change the thickness of hMMTs at early differentiation time points (days 0–4), but that as the hMMTs further matured, there was a bifurcation and TGFBR1 inhibitor-treated hMMTs were significantly thicker than control hMMTs. In this system, hMMT thickening was due to an increase in the width of individual fibers (Fig. 7d, e). Notably, SB-431542-treated human muscle fibers contained more nuclei than control fibers, confirming increased muscle cell fusion as the underlying cellular mechanism (Fig. 7f). In parallel, we ensured that TGFβ signaling does not interact with the anabolic Akt/mTOR anabolic pathway that drives myotube hypertrophy[38] in differentiated muscle cells (Fig. S6). Lastly, by treating hMMTs with acetylcholine (ACh) to induce tissue contraction and capturing short videos to visualize the magnitude of vertical rubber post deflections, we found that TGFBR1 inhibition renders the hMMTs stronger than their control counterpart (Fig. 7g, h). Together, these data demonstrate that TGFβ signaling regulates human cell fusion.

**Pharmacological modulation of TGFβ pathway in vivo perturbs muscle regeneration.** To evaluate the impact of TGFβ signaling on muscle cell fusion in vivo, we injured TA muscle of adult mice and injected either TGFβ1 protein or ITD-1 compound at 3 d.p.i., the time point when fusion begins (Fig. 8a). Evaluation of regenerating tissues at 7 d.p.i. revealed that both treatments lead to striking changes in myofiber size and morphology (Fig. 8b). TGFβ1 addition resulted in a robust decrease in nuclear number in newly formed myofibers (Fig. 8e), resulting in a dramatic drop in fiber cross-sectional area (Fig. 8c, d). In contrast, ITD-1 induced a large increase in myonuclear accretion (Fig. 8e) resulting in the formation of larger fibers (Fig. 8c, d). To further elucidate if modulating TGFβ signaling in vivo affects regenerated muscle tissue structure and function, we performed TA muscle injury, followed by three successive injections of TGFβ1 or ITD-1, and evaluated the regenerated muscles 2 weeks after injury (Fig. 8f). In this setup, the effects of modifying TGFβ signaling were more pronounced (Fig. 8g). Activation of TGFβ signaling induced the formation of very small myofibers, while ITD-1 treatment generated giant myofibers (Fig. 8h, i). We next performed in situ force measurement of regenerated TA muscles. As suggested by the severe myofiber atrophy observed in TGFβ1-injected muscles, ectopic activation of TGFβ signaling during tissue regeneration lead to a strong reduction of muscle-specific force (Fig. 8j). Despite being composed of larger myofibers, ITD-1-treated muscles did not show any improvement in force generation. These observations indicate that TGFβ signaling determines the numbers of fusion events occurring during tissue regeneration in vivo.

**Expression of actin-related genes is controlled by TGFβ signaling.** To identify the genetic networks regulated by TGFβ signaling in muscle cells, we performed transcriptome analysis of primary myocytes differentiated for 24 h and stimulated with either TGFβ1 or ITD-1 for another 24 h (Supplementary Data 1). Analysis of the transcriptomic dataset revealed that the relative expression levels of known TGFβ target genes (such as *Fibronectin*, *Lysyl oxidase*, etc.) were strongly modulated by either treatment, while the expression levels of myogenic transcription factors (such as *Pax7*, *Myod1*, etc.) were not significantly changed, confirming that the modulation of TGFβ signaling does not act

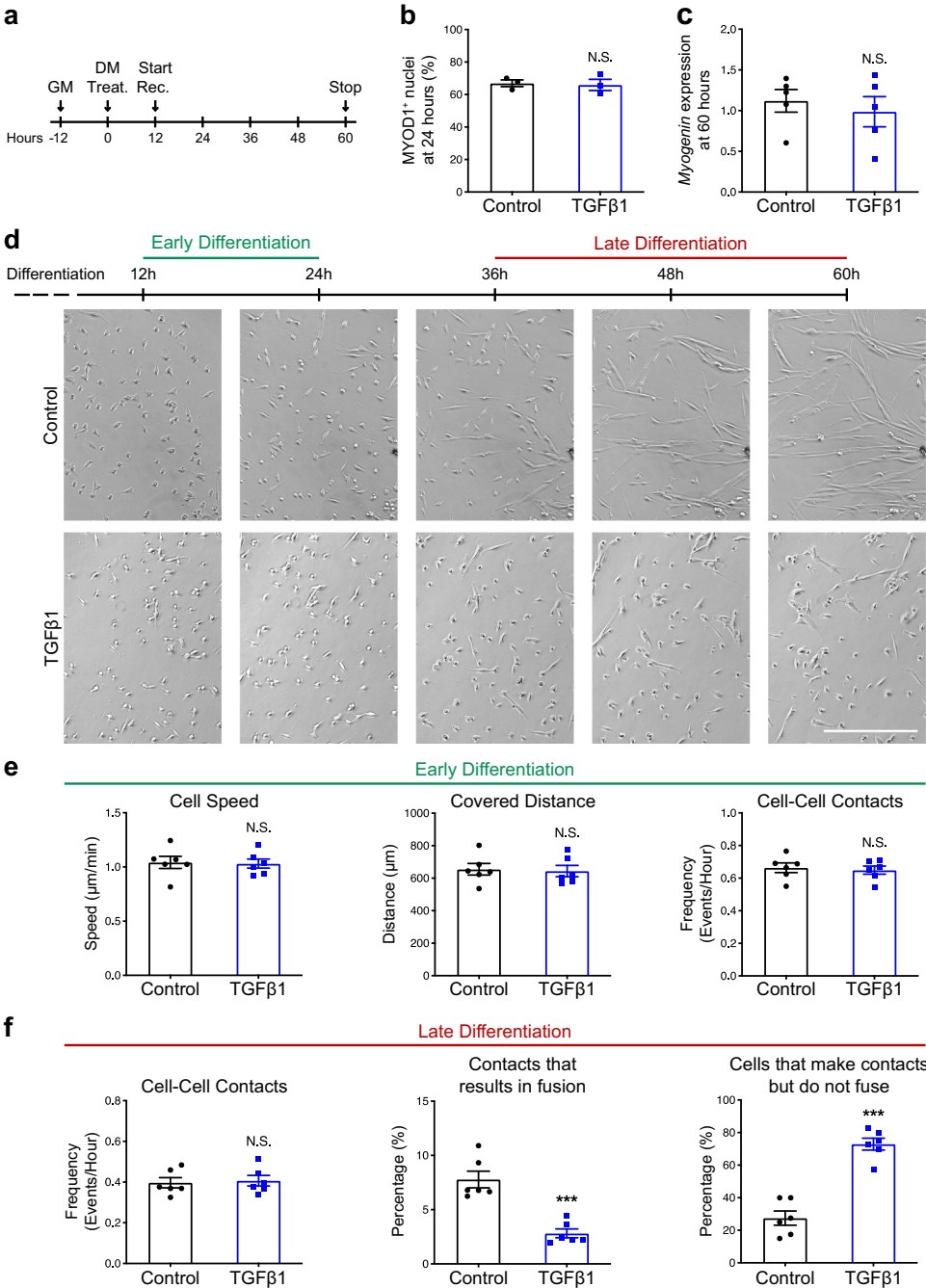

**Fig. 4 TGFβ signaling does not affect cell motility and cell–cell contact frequency. a** Experimental scheme. Primary myoblasts seeded at 10,000 cells/cm² were induced to differentiate with or without TGFβ1 recombinant protein. After 12 h, cells were recorded live for 12 h during early differentiation and for 24 h during late differentiation. **b** Quantification of the percentage of MYOD1+ nuclei of primary myocytes cultured for 48 h with or without TGFβ1 recombinant protein. N = 3 biologically independent experiments. **c** qRT-PCR analysis for *Myogenin* transcript expression of primary myotubes cultured for 60 h with or without TGFβ1 recombinant protein. N = 5 biologically independent experiments. **d** Brightfield live-imaging frames of differentiating myoblasts confirm that TGFβ stimulation reduces fusion. N = 6 primary cell cultures. **e** Early differentiation movies were used to quantify cell speed, covered distance and cell–cell contact frequency. None of these parameters were significantly different between control and TGFβ1-treated myoblasts. N = 240 cells for each condition examined over six biologically independent experiments. **f** Late differentiation movies were used to quantify cell–cell contact frequency, the percentage of cell–cell contacts that result in fusion events and the percentage of cells that make contacts but do not fuse. Although TGFβ1 stimulation does not impair cell–cell contact frequency, many TGFβ1-treated contacting myoblasts make contacts but do not accomplish fusion. N = 240 cells for each condition examined over six biologically independent experiments. Scale bars: **d** 400 μm. Data are presented as mean ± SEM. Unpaired two-tailed Student's *t* tests were used to compare between data. *** denote a significant difference with the Control group of P < 0.0001. NS not significant. Source data are provided as a Source Data file.

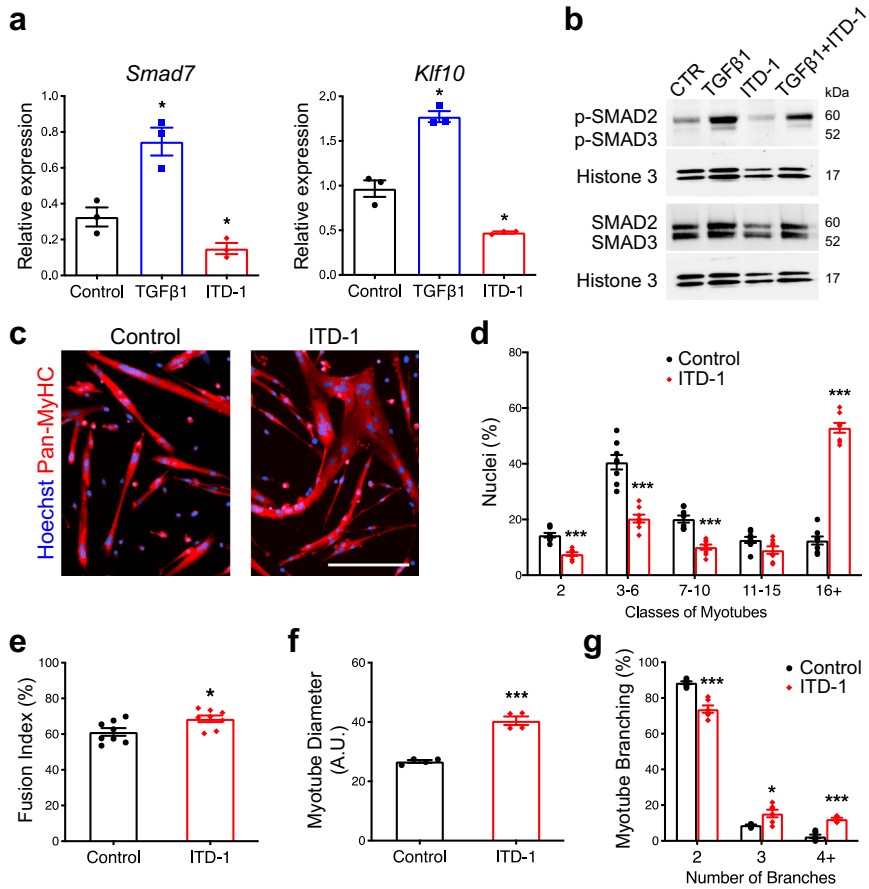

**Fig. 5 Inhibition of TGFBR2 function in differentiated muscle cell enhances fusion. a** qRT-PCR analysis of TGFβ target genes transcript expression in primary myocytes treated with TGFβ1 protein or ITD-1 compound proves that *Smad7* and *Klf10* are over-expressed when the signaling pathway is activated and inhibited when TGFβ cascade is blocked. $N = 10$ (Control), 8 (TGFβ1), and 8 (ITD-1) biologically independent experiments. **b** Nuclear p-SMAD2/3 and SMAD2/3 western blot analysis of primary myoblast treated with TGFβ1 protein, ITD-1 compound, or both combined. The intracellular mediators SMAD2/3 are phosphorylated upon TGFβ stimulation, while ITD-1 is able to reduce their phosphorylation. $N = 3$ biologically independent experiments. **c** Immunofluorescent staining for Pan-MyHC of re-plated myocytes cultured for 48 h. $N = 8$ primary cell cultures. **d** Aggregation index of re-plated myocytes shows that ITD-1 treatment leads to the formation of myotubes with higher numbers of nuclei compared to the control. $N = 8$ biologically independent experiments. **e** Fusion index of re-plated myocytes confirms the enhanced fusion when TGFβ cascade is inhibited. $N = 8$ biologically independent experiments. Diameter of re-plated myotubes (**f**) and of the distribution of branched-myotubes (**g**) of re-plated cells highlight aberrant morphology of syncytia treated with ITD-1 $N = 4$ (**f**) and 6 (**g**) biologically independent experiments. Scale bars: **c** 200 μm. Data are presented as mean ± SEM. Unpaired two-tailed Student's *t* tests were used to compare between data. * and *** denote a significant difference with Control group of $P < 0.05$ and $P < 0.001$, respectively. Source data are provided as a Source Data file.

on myogenesis in these experimental conditions. Likewise, recently identified fusion master regulatory factors (*Myomaker* and *Myomerger*) were also unaffected by TGFβ signaling (Fig. 9a). We then used Ingenuity Pathway Analysis (IPA) to reveal the pathways affected by TGFβ signaling. Interestingly, we found that "Actin Cytoskeleton" was among the top-regulated pathways (Fig. 9b). This is significant, since actin remodeling and the formation of finger-like actin protrusions are essential for myoblast fusion[39]. IPA further revealed changes in the transcription of numerous genes implicated in actin dynamics following TGFβ treatment (Supplementary Fig. 7a). Visualization of the F-ACTIN network and measurements of the local orientation of actin filaments in differentiated muscle cells showed that the level of TGFβ signaling negatively correlates with cytoskeleton reorganization (Fig. 9c). TGFβ signaling inhibited cell spreading (Fig. 9d) and the coherency of actin filament alignment (Fig. 9e). Importantly, the increased cell size in ITD-1-stimulated cells did not result in any changes in the frequency of cell–cell contacts (Fig. 9f).

To better understand actin remodeling during muscle cell fusion, we performed live-imaging experiments to visualize the accumulation of F-ACTIN foci in invasive podosome-like structures at the sites of fusion between cells[40]. By mixing cells stained with SiR-Actin and unstained cells, we observed dynamic actin remodeling in untreated cells (Supplementary Fig. 7b, top) and cells stimulated with ITD-1 (Supplementary Fig. 7b, Bottom). TGFβ1 stimulation prevented the formation of actin-rich invasive structures and promoted the maintenance of a rounded cell shape (Supplementary Fig. 7b, middle). We then asked if the fusogenic effect of ITD-1 can be suppressed by treating cells with Latrunculin, which prevents actin polymerization (Fig. 9g). We observed that Latrunculin treatment resulted in a decrease in fusion index (Fig. 9h), increased the proportion of small myotubes in both control and ITD-1-treated cultures (Fig. 9i), and completely prevented the formation of large myotubes containing high numbers of nuclei in ITD-1-treated cultures (Fig. 9j). These data demonstrate that inhibition of TGFβ signaling does not promote fusion in cells that are capable of fusion, but have perturbed actin dynamics.

To gain more mechanistic insights into the regulation of fusion by TGFβ, we followed a candidate–gene approach among the

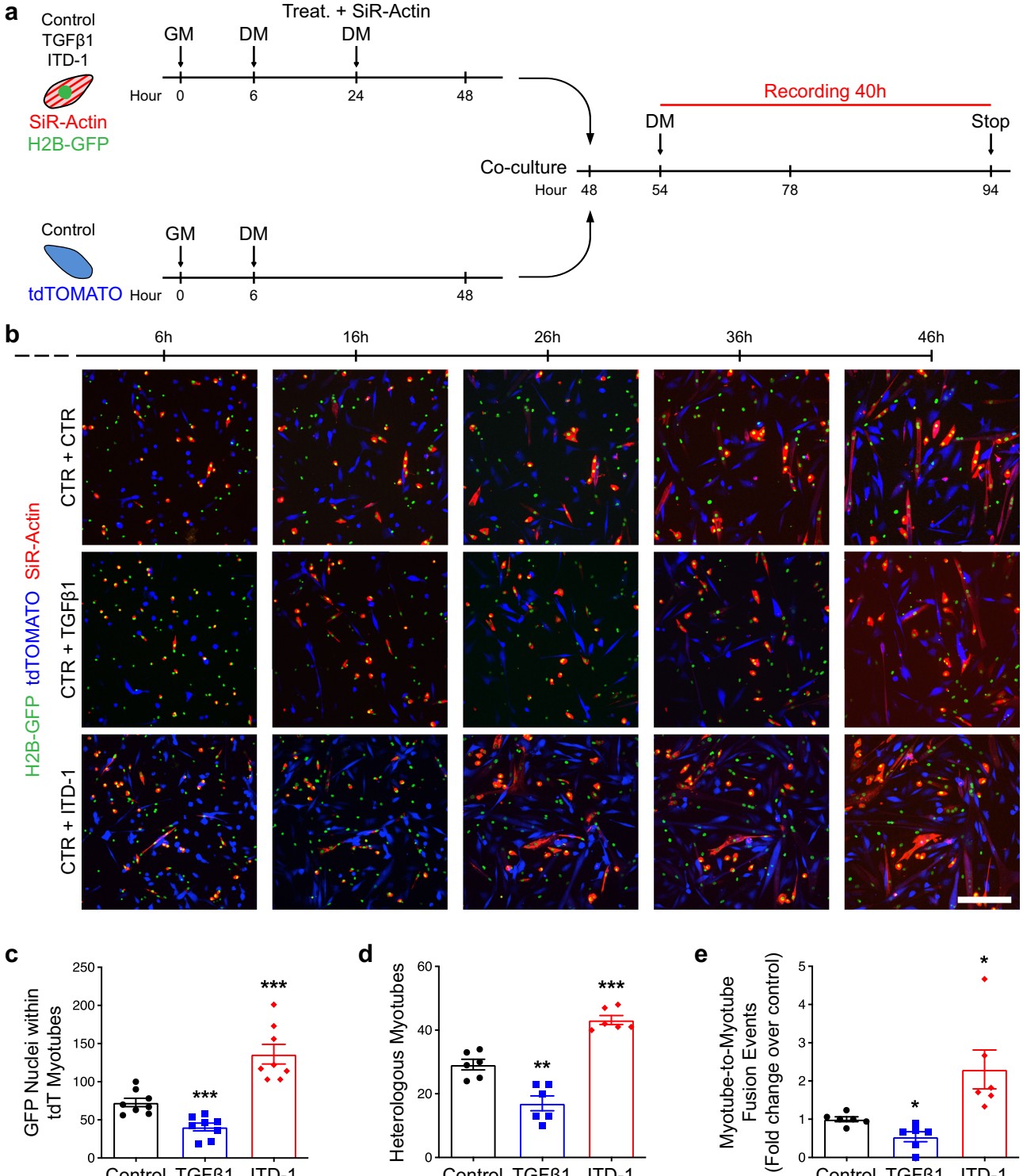

**Fig. 6 Live imaging of myoblast fusion. a** Experimental scheme. H2B-GFP primary myoblasts were seeded at low density (5000 cells/cm$^2$), treated with TGFβ1 protein or ITD-1 compound, stained with SiR-Actin, and differentiated for 2 days. Membrane-tdTOMATO primary myoblasts seeded at low density (5000 cells/cm$^2$) and were differentiated for 2 days. Both populations were split and co-cultured (50/50) at high density (75,000 cells/cm$^2$) for 2 more days. In the last 40 h, cells were recorded live by confocal microscopy. **b** Live-imaging frames of co-cultured pre-differentiated myocytes confirm the phenotype previously observed. TGFβ activation inhibits fusion, while ITD-1 enhance fusion. $N = 8$ biologically independent primary co-cultures. **c** Quantification of H2B-GFP nuclei within tdTOMATO myotubes. $N = 8$ biologically independent experiments. **d** Quantification of heterologous myotubes (double positive for SiR-Actin and tdTOMATO). $N = 6$ biologically independent experiments. **e** Quantification of Myotube-to-Myotube events. ITD-1 treatment allows more myotube-to-myotube events compared to the control. $N = 6$ biologically independent experiments. Scale bars: **b** 200 μm. Data are presented as mean ± SEM. Unpaired two-tailed Student's $t$ tests were used to compare between data. *, **, and *** denote a significant difference with the Control group of $P < 0.05$, $P < 0.01$, and $P < 0.001$ respectively. Source data are provided as a Source Data file.

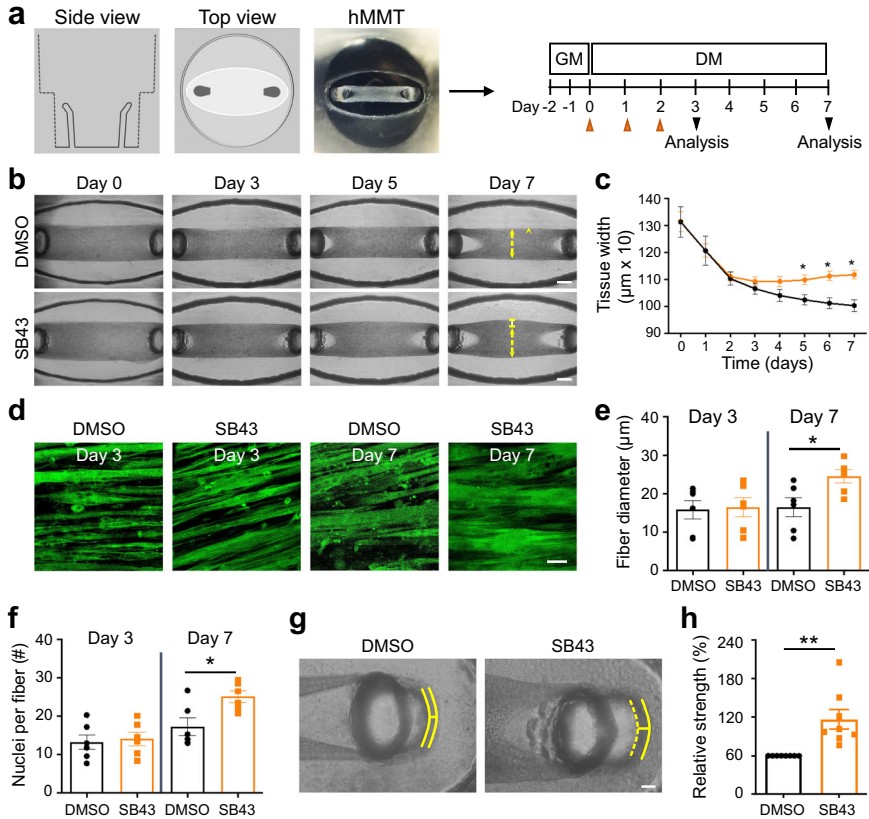

**Fig. 7 TGFβ inhibition induces human myotube fusion in 3D culture resulting in increased microtissue strength. a** Schematic representation (left) and timeline (right) of 3D human muscle cell experimental approach utilized in panels (**b**–**h**). Briefly, immortalized human myoblasts are suspended in a fibrin/reconstituted basement membrane protein scaffold and seeded into the bottom of a custom rubber 96-well plate culture device. A side view depicts the vertical posts across which the cells remodel the protein scaffold, align, and fuse to form a 3D human muscle microtissue (hMMT). For the first 2 days of culture (Day −1, Day −2), tissues are maintained in growth media (GM). On Day 0, GM is removed from wells and replaced with differentiation media (DM). TGFBR1 inhibitor SB-431542 (SB43, 10 μM) was included in the DM on Days 0–2 (orange arrowheads) of culture. **b** Representative bright-field images of 3D hMMT culture over the time course of differentiation treated with SB43 as compared to DMSO-treated control. White arrows demarcate the region of tissues that are assessed in panel (**c**). **c** Line graph quantifying hMMT width over the time course of differentiation in DMSO (black line) or SB43 (orange line) conditions. $N = 14$ (DMSO) and 15 (SB43) biologically independent experiments. **d** Representative confocal slices of hMMT cultures immunostained for SARCOMERIC α-ACTININ (green) on Days 3 and 7 of culture. **e, f** Bar graph quantifying muscle fiber diameter (**e**) and average number of nuclei per fiber (**f**) at Days 3 and 7 of culture. $N = 6$ biologically independent experiments for each time point. **g** Representative brightfield images of hMMTs. Micro-post position before (solid yellow line) and after (dashed yellow line) acetylcholine stimulation is represented. **h** Bar graph quantifying relative strength of SB43-treated hMMTs compared to DMSO-treated hMMTs. $N = 8$ biologically independent experiments. Scale bars: **b**, 500 μm; **d** 50 μm; **g** 100 μm. A minimum of 30 microscopic images per culture condition was analyzed. Data are presented as mean ± SEM. Unpaired two-tailed Student's *t* tests were used to compare between data. * and ** denote a significant difference with Control group of $P < 0.05$ and $P < 0.01$, respectively. Source data are provided as a Source Data file.

transcripts with expression most impacted by both TGFβ1 and ITD-1. We selected *Timp1* (induced by TGFβ, Supplementary Fig. 8b), an inhibitor of matrix metalloproteinases, which are known to promote actin-rich structures in invadopodias[41], and *Arhgef6* (inhibited by TGFβ, Supplementary Fig. 8e), which acts as a RAC1 guanine nucleotide exchange factor (GEF)[42], a crucial regulator of actin cytoskeleton. We observed that either exogenous TIMP1 protein addition (Supplementary Fig. S8c) or small interfering RNA (siRNA)-mediated silencing of *Arhgef6* expression (Supplementary Fig. 8f) slightly reduced the fusion of muscle cells, without impacting their ability to differentiate. Taken together, our data indicate that TGFβ signaling limits cell fusion by preventing actin cytoskeleton remodeling.

Lastly, we asked whether the TGFβ-driven effect on fusion is conserved in a non-muscle context. To do so, we took advantage of a fibroblast dox-inducible cell fusion reconstitution system[43]. Myomaker- and Myomerger-transduced 10T1/2 fibroblasts were

seeded and stimulated with dox to induce fusion (Supplementary Fig. 9a). Recombinant TGFβ1 recombinant protein was administered at multiple time points (from days 0 to 3), but none of these treatments led to a reduction of fusion (Supplementary Fig. 9b, c). These results show that TGFβ1 protein is unable to inhibit the fusion process in a non-muscle cell type and that its signaling cascade is unable to physically prevent fusion through Myomaker and Myomerger, confirming the evidence obtained from the transcriptome analysis (Fig. 9a). Finally, our data suggest that TGFβ acts on the fusion process before the final step mediated by Myomaker/Myomerger.

## Discussion
The data presented here identify an unexpected negative role for TGFβ in the fusion of adult myocytes to form myotubes. TGFβ signaling has previously been shown to play a major role in

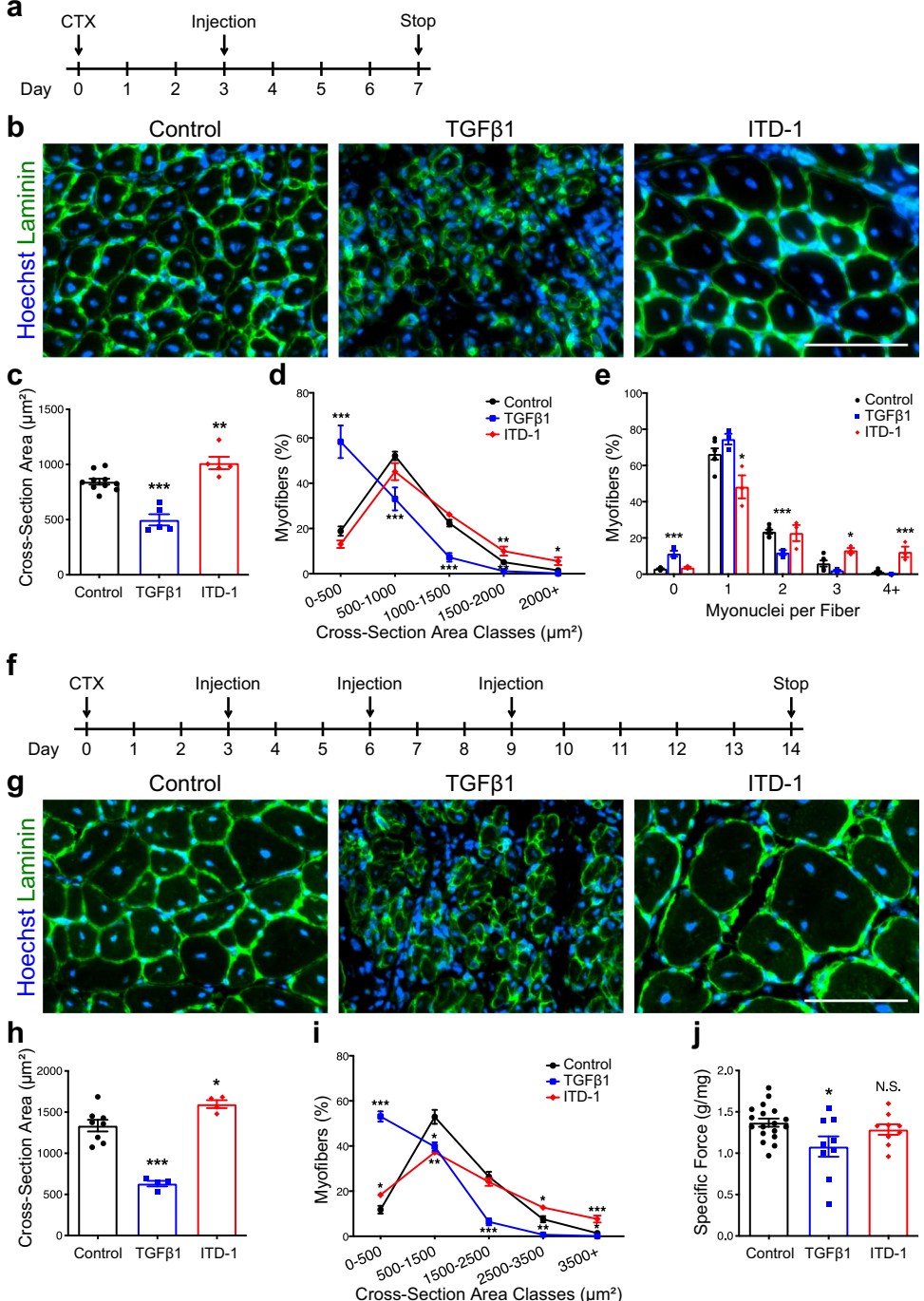

**Fig. 8 TGFβ signaling regulates muscle cell fusion in vivo. a** Experimental scheme. Adult murine *tibialis anterior* (TA) muscles were subjected to CTX injury and regenerating tissues were injected intramuscularly with either TGFβ1 proteins or ITD-1 compound 3 days after damage. **b** Immunofluorescent staining for LAMININ of 7 days regenerating TA muscles. **c** Quantification of myofiber size (cross-sectional area, CSA). While the injection of TGFβ strongly reduces fibers size, ITD-1 administration increases fibers size. **d** Distribution of myofiber CSA. N = 10 (Control), 5 (TGFβ1), and 5 (ITD-1) biologically independent TA muscles. **e** Distribution of myonuclei per fiber shows that the inhibition of TGFβ cascade leads to the formation of multinucleated myofibers, while TGFβ activation reduces the number of myonuclei per fibers. N = 6 (Control), 3 (TGFβ1), and 3 (ITD-1) biologically independent TA muscles. **f** Experimental scheme. Adult murine TA muscles were subjected to CTX injury and regenerating tissues were injected with either TGFβ proteins or ITD-1 compound 3, 6, and 9 days after damage. Fourteen days after injury, force measurements were performed, and TA muscles were collected. **g** Immunofluorescent staining for LAMININ of 14-day regenerating TA muscles. **h** Quantification of myofiber size confirms the phenotypes observed at 7 d.p.i. N = 8 (Control), 4 (TGFβ1), and 4 (ITD-1) biologically independent TA muscles. **i** Distribution of myofiber CSA. N = 8 (Control), 4 (TGFβ1), and 4 (ITD-1) biologically independent TA muscles. **j** Specific force measurement of regenerating muscles. While TGFβ1-treated muscles are weaker compared to the control, ITD-1-injected muscles show no differences. N = 18 (Control), 9 (TGFβ1), and 9 (ITD-1) biologically independent TA muscles. Scale bars: **b**, **g**, 100 μm. Data are presented as mean ± SEM. Unpaired two-tailed Student's *t* tests were used to compare between data. *, **, and *** denote a significant difference with control group of P < 0.05, P < 0.01, and P < 0.001, respectively. Control represents mock-treated contralateral TA muscle. Source data are provided as a Source Data file.

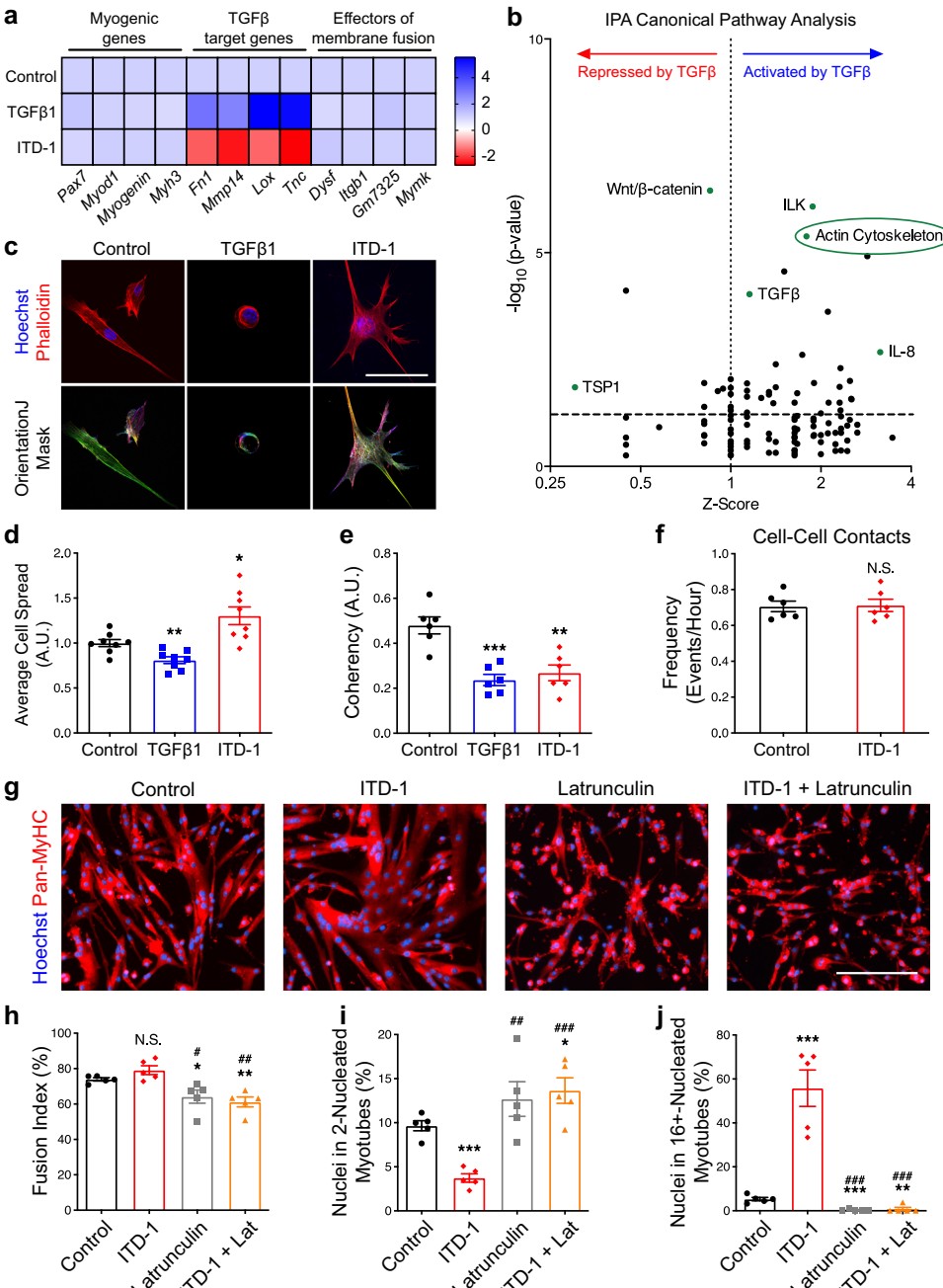

**Fig. 9 Fusogenic actin remodeling is controlled by TGFβ signaling.** Transcriptomic analysis was performed on differentiated myocytes treated with either TGFβ1 or ITD1. $N = 3$ biologically independent experiments. **a** Heatmaps of TGFβ target genes, myogenic genes, and fusion genes. **b** Volcano plot showing the Ingenuity Pathway Analysis (IPA). Among the top modulated pathways, Actin Signaling Pathway is highlighted. **c** Phalloidin staining of 1-day differentiated myocytes. These pictures were analyzed with OrientationJ (ImageJ Plug-in) to obtain a color-coded orientation mask. $N = 8$ primary cell cultures. **d** Average cell spread quantification. TGFβ1 treatment reduces cell size; ITD-1 promotes cell spreading. $N = 8$ primary cell cultures. **e** Quantification of orientation coherency of the actin fibers. Both treatments reduce coherency compared to the control. $N = 150$ cells for each condition examined over six biologically independent experiments. **f** Quantification of cell–cell contact frequency. $N = 240$ cells for each condition examined over six biologically independent experiments. **g** Immunofluorescent staining for Pan-MyHC of re-plated primary myotubes cultured for 48 h with ITD-1, Latrunculin, or both. $N = 5$ primary cell cultures. **h** Fusion index of re-plated myotubes shows that Latrunculin significantly reduces the parameter when administrated. $N = 5$ biologically independent experiments. **i** Percentage of nuclei in the smallest myotube classes. ITD-1-treated myotubes are characterized by a lower number of nuclei in the smallest myotubes, while Latrunculin increases the percentage of nuclei in small myotubes when administrated alone or together with ITD-1. $N = 5$ biologically independent experiments. **j** Percentage of nuclei in the biggest myotube classes. ITD-1 strongly increases the number of nuclei in big myotubes, but Latrunculin blunts this effect, reducing the percentage. $N = 5$ biologically independent experiments. Scale bars: **c** 40 μm; **g** 200 μm. Data are presented as mean ± SEM. Unpaired two-tailed Student's $t$ tests were used to compare between data. *, **, and *** denote a significant difference with the Control group of $P < 0.05$, $P < 0.01$, and $P < 0.001$ respectively. #, ##, and ### denote a significant difference with ITD-1 group of $P < 0.05$, $P < 0.01$, and $P < 0.001$, respectively. NS not significant. Source data are provided as a Source Data file.

skeletal muscle morphogenesis. Throughout development, TGFβ ligands are expressed mostly by connective tissue cells and in close proximity to growing muscle tissue[44]. While it is known that connective tissue cells provide a pre-pattern for limb muscle patterning[45] and control the amount of myofibers within the developing muscle masses[46], we propose that TGFβ is the main signal limiting muscle cell fusion. As such, TGFβ could regulate muscle homeostasis and the proper shape of myofibers.

During adult tissue repair, TGFβ1 and TGFβ3 are mainly expressed by inflammatory macrophages invading the regenerating muscle tissue[47,48], while TGFβ2 is secreted by activated satellite cells and differentiating myotubes[47]. This sequential expression of ligands by different cell types may be instrumental in preventing the premature fusion of transient amplifying myoblasts, inappropriate fusions between syncytia, and the formation of aberrant branched myotubes. Indeed, we speculate that it is the lack of other cell types in 3D human skeletal muscle microtissue that stunts the maturation of myotubes. It is only upon introducing a missing signal to induce myotube-to-myotube fusion that we release a developmental brake that prevents the next phase of tissue maturation.

Moreover, we found that proliferating primary muscle cells express both TGFβ1 and TGFβ2. Interestingly, while the expression levels of these two isoforms dramatically decrease during the differentiation/fusion process, TGFβ3 expression is increased. Since we observed that p-SMAD3 signaling is reduced but not abrogated in fusing muscle cells, we propose that TGFβ3 is instrumental in maintaining a basal level of TGFβ signaling activity to prevent excessive fusion. The persistence of p-SMAD3 signaling in newly formed (centrally-nucleated) myofibers during muscle tissue regeneration suggest that TGFβ signaling prevents additional cells from fusing into these still immature myofibers. This concept is strengthened by our observation that inhibition of remaining TGFβ signaling enhances fusion. TGFβ3 may be an autocrine signal for myotubes to keep fusion "in check."

In the context of disease, myofibers with branches are found in muscular dystrophy[49]. They present morphological malformations, as well as alterations in calcium signaling[50], and arise from asynchronous myofiber remodeling. Aberrant fusion events driven by chronic elevation of TGF-β signaling in muscle pathologies[51], associated with impaired regeneration, may contribute to disease progression and the severity of disease phenotypes. Knowledge of the signaling pathways regulating muscle cell fusion may help design therapeutic strategies to decrease myofiber branching in dystrophic patients.

Our in vivo experiments indicate that TGFβ signaling must be tightly regulated in muscle progenitor cells during tissue repair. Treatment of regenerating muscle tissues with TGFβ1 protein strongly blocks muscle progenitor cell fusion and impairs the function of regenerated tissue. Interestingly, ITD-1 administration leads to the formation of giant myofibers containing more nuclei. However, while functional, the regenerated muscle did not generate higher force compared to mock-treated regenerated tissues. While appealing, it remains to be demonstrated if enhancing cell fusion might improve regenerated muscle function. As such, "bigger" does not always means "stronger", and this is exemplified by our previous analysis showing that lack of Rspo1 results in the formation of larger myofibers containing supernumerary nuclei following regeneration[6]. Further to that point, the end goal of coordinated muscle tissue regeneration is to restore a functional tissue architecture and mechanical properties in accordance with the other components of the musculoskeletal system.

Work in *Drosophila melanogaster* previously demonstrated that actin polymerization drives muscle cell fusion[39]. Our demonstration that TGFβ stimulation breaks down actin architecture link extracellular cues to cell mechanics. We show that the state of TGFβ signaling in pre-fusing cells controls their shape, and the formation of actin-based protrusions that are necessary for the fusion of mammalian cells[40]. Importantly, TGFβ signaling may regulate multiple points of the fusion pathway through actin nucleation. Our results also demonstrate that the blockade of actin polymerization blunts the over-fusion phenotype induced by inhibition of TGFβ signaling. Interestingly, TGFβ stimulation of Myomaker- and Myomerger-expressing fibroblast enhanced fusion of these cells. This can be explained by the fact that TGFβ induces local cytoskeletal reorganization and promote F-ACTIN polymerization in fibroblasts[52,53] in the opposite way as we observed in muscle cells. Future work should investigate why TGFβ signaling has a different effect on actin dynamics, depending on the cell identity.

In the present state of our knowledge, most of the fusion-promoting factors have been discovered through in vivo studies in the fly embryo. As such, the concept of the fusogenic synapse, the site where an attacking fusing cell propels an actin-rich membrane protrusion towards a receiving cells, is poorly characterized in mammalian cells[54]. Here, we identify numerous actin-related transcripts that are regulated by TGFβ signaling and may be integral parts of the molecular fusion machinery. While our data hints that TIMP1 and ARHGEF6, among others, may be effectors of TGFβ signaling at the level of cell fusion, further work should thus be dedicated to the study of the TGFβ-regulated genes, and their specific roles in actin dynamics to increase our understanding of how muscle cell membranes are brought together for fusion.

In conclusion, the elucidation of TGFβ signaling as a brake for myoblast fusion opens new avenues to study this fundamental cellular process at a molecular level and to understand how fusion is perturbed in neuromuscular diseases. Our demonstration that blockade of TGFβ signaling enhances human cell fusion may lead to new therapeutic approaches for pathologies associated with defects in muscle stem cell fusion to their host myofibers.

## Methods

**Mice**. Wild-type (WT) mice used in this project were 2 to 5 months-old C57Bl6/N mice purchased from Janvier Laboratories. Experiments were performed at the Centre d'Expérimentation Fonctionnelle (UMS28) Animal Facility following the European regulations for animal care and handling. Experimental animal protocols were performed in accordance with the guidelines of the French Veterinary Department and approved by the Sorbonne Université Ethical Committee for Animal Experimentation. Cardiotoxin (CTX) injection in TA muscle and hindlimb muscle dissection were performed following the protocol described in ref. [55].

**Skeletal muscle injury**. Mice were anesthetized by intraperitoneal injection of ketamine at 0.1 mg/g body weight and xylazine at 0.01 mg/g body weight diluted in saline solution. 30 μl of CTX (12 mM in saline, Latoxan) was injected into hindlimb TA muscles to induce injury, and mice were euthanized 0, 1, 2, 3, 4, 5, 7, or 14 days afterward. Recombinant mouse TGFβ1 (R&D Systems) was diluted in saline and 250 ng (25 μl) was injected into the TA every injection. ITD-1 compound was diluted in saline and 2000 ng (25 μl) was injected into the TA every injection. Muscles were freshly frozen in OCT Embedding Matrix compound (CellPath) and cut transversally at 10 μm with a Leica cryostat.

**In situ physiological assay**. TA muscles were evaluated by the measurement of in situ isometric muscle contraction in response to nerve stimulation. Mice were anesthetized intraperitoneal injection of ketamine at 0.1 mg/g body weight and xylazine at 0.01 mg/g body weight diluted in saline solution. Feet were fixed with clamps to a platform and knees were immobilized using stainless-steel pins. The distal tendons of muscles were attached to an isometric transducer (Harvard Bioscience) using a silk ligature. The sciatic nerves were proximally crushed and distally stimulated by bipolar silver electrode using supramaximal square wave pulses of 0.1 ms duration. All data provided by the isometric transducer were recorded and analyzed on a microcomputer, using PowerLab system (4SP, AD Instruments). All isometric measurements were made at an initial length $L_0$ (length at which maximal tension was obtained during tetanus). Responses to tetanic stimulation (pulse frequency from 75 to 143 Hz) were successively recorded. The

maximal tetanic force was determined. Muscle masses were measured to calculate specific force.

**Murine cell cultures.** Skeletal muscle-derived primary myoblasts were isolated from WT mice using the Satellite Cell Isolation Kit MACS protocol (Miltenyi Biotec). Briefly, hindlimb muscles were dissected out, placed in a sterile Petri dish, and minced to a pulp with a curved scissor. The pulp was then incubated in a CollagenaseB/DispaseII/CaCl$_2$ solution at 37 °C for 40 min with two trituration steps. The enzymes are then blocked by the addition of fetal bovine serum (FBS) and the muscle extract was treated with Red Blood Cell Lysis Solution to remove erythrocytes. After this step, magnetic labeling is performed by adding Buffer (5% bovine serum albumin (BSA), 2 mM EDTA, phosphate-buffered saline (PBS)) and Satellite Cell Isolation Kit (a mixture of antibodies specific for non-satellite cells conjugated with magnetic beads). The cell suspension is then poured into the column in the magnetic field. Unlabeled cells (satellite cells) flow through the column while magnetically labeled cells are retained within the column. Satellite cells were resuspended in the growth medium (Ham's F10 with 20% FBS, 1% penicillin/streptomycin (P/S), and 2.5 ng/ml of basic fibroblast growth factor) and plated into a collagen-coated 60 mm Petri dish. Cells were maintained in the growth medium until cells reached 80% confluence. To induce myogenic differentiation and fusion, myoblasts were plated at different concentrations depending on the experimental design (5000, 20,000, or 75,000 cells/cm$^2$) onto Matrigel-coated plates in the growth medium. Once adherent, cells were incubated in the differentiation medium (Dulbecco's modified Eagle's medium (DMEM) with 2% horse serum and 1% P/S) for up to 3 days.

**Cell treatments and RNA interference.** For recombinant protein treatments, TGFβ1 (Thermo Fisher Scientific), TGFβ2 (R&D System), and TGFβ3 (R&D System) were administered at a final concentration of 20 ng/ml. ITD-1 (R&D System) and SB-431542 (Sigma-Aldrich) compounds were administered at a final concentration of 5 μM. TIMP1 (R&D System) recombinant protein was used at a final concentration of 500 ng/ml. Latrunculin B (Santa Cruz BioTechnology) was administered at a final concentration of 500 nM. For RNA interference experiments, primary cells were seeded in collagen-coated plate in antibiotic-free growth medium. siRNA transfection was performed using Lipofectamine 2000 and Opti-MEM according to the manufacturer's protocol. Both siArhgef6 (Thermo Fisher Scientific; Cat#4390771; ID: s91693) and siControl (Thermo Fisher Scientific; Cat#4390844) were used at a 33 mM concentration.

**Polydimethylsiloxane (PDMS) mold fabrication for 3D hMMTs.** To generate 3D hMMTs, we employed a second-generation micro-molded device in a 96-well format made of PDMS (monomer/cross-linker ratio = 15:1) in a single simple molding process[56]. At the bottom of each well of the 96-well microfabricated device, an oval-shaped pool was designed with a vertical flexible PDMS post on each side of it. PDMS culture plates were sterilized via an autoclave. Just prior to use, wells were further sterilized by a overnight incubation with a 5% pluronic acid solution (100 μl/well) at 4 °C, which also served to create a non-adhesive surface to support tissue self-organization.

**hMMT culture.** The 3D hMMTs were generated using human immortalized myoblast lines obtained from V. Mouly (AB1167 from fascia lata muscle of a healthy 20-year-old male, AB1190 from the paravertebral muscle of a healthy 16-year-old male, and KM155 from thigh muscle of a healthy 25-year-old male)[57]. Immortalized myoblasts were cultured in Skeletal Muscle Cell Basal Medium with Skeletal Muscle Cell Growth Medium Supplement Mix (PromoCell) supplemented with 20% FBS and 1% P/S. Myoblasts were harvested by trypsinization and resuspended (1.5 × 10$^5$ cells/tissue or 1.0 × 10$^7$ cells/ml) in a hydrogel mixture consisting of fibrinogen (4 mg/ml, 40% v/v; Sigma-Aldrich) and Geltrex (20% v/v; Thermo Fisher Scientific) in DMEM (40% v/v) in the absence of thrombin. Then, 0.2 U of thrombin (Sigma-Aldrich) per each mg of fibrinogen was added just before seeding the cell–hydrogel mixture into the wells and left for 5 min in an incubator at 37 °C to allow optimal fibrin gel formation. Subsequently, 200 μl of growth medium consisted of Skeletal Muscle Cell Growth Medium lacking supplement mix (Skeletal Muscle Cell Basal Medium; PromoCell) supplemented with 20% FBS, 1% P/S, and 1.5 mg/ml 6-aminocaproic acid (ACA; Sigma-Aldrich) was added to each well. The hMMTs were cultured in the growth medium for 2 days, and then the medium was replaced with the differentiation medium (DMEM supplemented with 2% horse serum, 1% P/S, and 10 μg/ml human recombinant insulin) containing 2 mg/ml ACA to induce differentiation. SB-431542 (Sigma-Aldrich) was added at a final concentration of 10 μM into the differentiation medium of hMMTs for Days 0–2 of differentiation, while an equivalent volume of DMSO was added to control samples. On Day 3 of differentiation, a final full media exchange was performed to remove SB-431542 or DMSO. Half of the culture medium was replaced every other day for the remaining differentiation period.

**3D tissue compaction and tissue remodeling analysis.** The effect of SB-431542 treatment on hMMT compaction was evaluated by measuring the tissue diameter as an indication for tissue self-organization and remodeling over the differentiation period. Phase-contrast ×4 magnification images were captured over time using an inverted microscope (Olympus) to analyze 3D tissue compaction. In each image, four-width measurements were done across the length of the tissue using ImageJ software and the average diameter was calculated. The data were shown as absolute diameter change over time and the result was compared between SB-431542- and DMSO-treated control tissues.

**hMMT myofiber width and nuclear index analyses.** Myofiber width was measured using ×40 magnification stack images of SB-431542- and DMSO-treated hMMTs at Days 3 and 7 of the differentiation period. Analysis of SAA immunostained images of 3D muscle tissues was facilitated using the NIH ImageJ software. We analyzed a total of three to five images per tissue, to determine the diameter of each muscle fiber. Myofibers were only qualified for fiber diameter analysis if they were visible across the length of the stacked image. In this work, myotubes were defined as multinucleated cells comprising at least three fused nuclei. Three-width measurements were done across the length of each qualified fiber to ensure that the thickest and the thinnest parts were included in the measurements, and, subsequently, the average fiber diameter per condition was calculated. To determine the average number of nuclei per fibers, we quantified the total number of Hoechst+ nuclei contained within each SAA+ muscle fiber in each hMMT culture condition.

**hMMT relative force quantification.** To evaluate the effect of TGFβ inhibition on the function of hMMTs, we evaluated hMMT contraction in response to ACh (Sigma-Aldrich) stimulation and tissue contraction. Briefly, ACh solution (in DMEM) was directly added (1 mM final concentration) into the wells containing hMMTs after 7 days of differentiation. We then captured short-phase-contrast videos at ×10 magnification to visualize the movement of the flexible PDMS posts. Post displacement was quantified using the ImageJ software. Relative hMMT strength data were evaluated by normalizing the post displacement of SB-431542-treated tissues to DMSO-treated control tissues.

**Myomaker- and Myomerger-expressing 10T1/2 fibroblasts.** Myomaker- and Myomerger-expressing fibroblasts were used and described previously[16]. Myomaker- and Myomerger-transduced 10T1/2 fibroblasts were seeded in 8-chamber Ibidi slides with a 3 × 10$^3$ cell density per well (Day 0). After 8 h of post seeding (Day 0), Myomerger expression was induced by treating the cells with dox-containing culture medium (1 μg/ml) and replaced every 24 h. Each experimental chamber was treated with human TGFβ1 recombinant protein (20 ng/ml; Thermo Fisher Scientific) as specified. Four days after seeding, cells were fixed and fusion was evaluated by analyzing the number of nuclei in GFP+ cells. The experiment was performed three times in duplicate and at least three images per well were quantified.

**RNA extraction and quantitative real-time PCR (qRT-PCR).** Total RNA was isolated from cultured cells and TA muscles using TRIzol Reagent (Thermo Fisher Scientific) or Direct-zol RNA Kit (Zymo Research) according to the manufacturer's protocol. TA muscle tissue was destroyed using the MagNa Lyser System (Roche). RNA concentration was evaluated with Nanodrop. After subsequent DNAse treatment, complementary DNA (cDNA) was generated using High Capacity Reverse Transcription Kit (Thermo Fisher Scientific). cDNA was then used for quantitative PCR (qPCR) done with LightCycler 480 SYBR Green Master Mix (Roche) and run in LC480 for 40 cycles. Primers are reported in Supplementary Table 1. All samples were duplicated, and transcripts levels were normalized for a housekeeping gene relative abundance (TBP, transcription regulator).

**Microarray and bioinformatics.** The RNA from primary myoblasts were isolated using Direct-zol RNA Kit (Zymo Research) according to the manufacturer's protocol. After validation of the RNA quality with Bioanalyzer 2100 (using Agilent RNA6000 Nano Chip Kit), 100 ng of total RNA is reverse transcribed following the GeneChip® WT Plus Reagent Kit (Affymetrix). Briefly, the resulting double-stranded cDNA is used for in vitro transcription with T7 RNA polymerase (all these steps are included in the WT cDNA synthesis and amplification kit of Affymetrix). After purification according to Affymetrix protocol, 5.5 μg of Sens Target DNA are fragmented and biotin labeled. After control of fragmentation using Bioanalyzer 2100, cDNA is then hybridized to GeneChip® MouseGene2.0ST (Affymetrix) at 45 °C for 17 h. After overnight hybridization, chips are washed on the fluidic station FS450 following specific protocols (Affymetrix) and scanned using the GCS3000 7G. The scanned images are then analyzed with the Expression Console software (Affymetrix) to obtain raw data (cel files) and metrics for Quality Controls. Data were normalized using RMA algorithm in Bioconductor with the custom CDF vs 22. Statistical analyses were carried out with the use of Partek® GS. First, variations in gene expression were analyzed using unsupervised hierarchical clustering and principal component analysis to assess data from technical bias and outlier samples. To find differentially expressed genes, we applied a one-way analysis fo variance for each gene. Then, we used unadjusted $P$ value and fold changes to filter and select differentially expressed genes. In TGFβ1- or ITD-1-treated myocytes, the genes were selected with $P < 0.05$ significance and at least 50% difference. Gene networks and canonical pathways representing key genes were identified using the curated IPA® database.

**Immunofluorescence**. Cell cultures were fixed with 4% paraformaldehyde in PBS for 15 min, washed three times with PBS, and permeabilized with 0.25% Triton X-100 in PBS for 10 min. The blocking step was performed using 4% BSA for 45 min at room temperature. Cultures were then incubated with primary antibodies diluted in blocking solution (4% BSA) overnight at 4 °C. After a quick wash with 0.1% NP40 in PBS, samples were washed with PBS twice. Cells were incubated with secondary antibodies for 45 min at room temperature. Secondary antibodies used were anti-mouse immunoglobulin G (IgG) or anti-rabbit IgG coupled with Alexa Fluor 488 or 546 dyes, from Thermo Fisher Scientific. Nuclei were counterstained with Hoechst (1:1000; Thermo Fisher Scientific). F-actin staining was performed using fluorescein isothiocyanate-conjugated Phalloidin (1:500; Sigma-Aldrich) or SiR-Actin (100 nM; Spirochrome). Following antibody staining, cultures were washed three times with PBS and nuclei were stained with Hoechst (1:1000; Thermo Fisher Scientific), before being analyzed with the EVOS FL Cell Imaging System microscope (Thermo Fisher Scientific) or with a Nikon Ti2 microscope equipped with a motorized stage and a Yokogawa CSU-W1 spinning disk head coupled with a Prime 95 sCMOS camera (Photometrics).

Cryosections were fixed with 4% paraformaldehyde (PFA) in PBS (Electron Microscopy Sciences), washed three times with PBS, and then incubated with methanol for 6 min at −20 °C. For Pax7 staining, cryosections were incubated with citric acid (0.1 M) for 10 min at 96 °C. For Myogenin staining, methanol incubation was not performed, and cryosections were incubated with Antigen Unmasking Solution (Vector) for 10 min at 96 °C. The blocking step was performed using 2% horse serum, 2% BSA, and 0.1% Triton X-100 PBS for 1 h at room temperature. After blocking, the cryosections were incubated overnight with primary antibodies diluted in blocking solution at 4 °C. Detection of primary antibodies was achieved using Alexa Fluor secondary antibody (Thermo Fisher Scientific) at a 1:1000 dilution in PBS. Nuclei were counterstained with Hoechst (1:1000; Thermo Fisher Scientific).

hMMT samples were fixed with 4% formaldehyde (Alfa Aesar) for 15 min at room temperature. Following three washes with PBS, samples were permeabilized and blocked with a blocking solution containing 10% goat serum (Thermo Fisher Scientific) and 0.3% Triton X-100 in PBS for 30 min at room temperature (RT). Samples were then incubated with a mouse anti-sarcomeric α-actinin antibody (Sigma-Aldrich) diluted 1:800 in the blocking solution overnight at 4 °C. After three washes with PBS, samples were then incubated with goat anti-mouse IgG conjugated with Alexa-Fluor 488 secondary antibody (1:500; Thermo Fisher Scientific) and Hoechst (1:1000; Thermo Fisher Scientific) diluted in the blocking solution for 60 min at RT. Multiple confocal stacks through each tissue were captured at multiple randomized locations using an Olympus IX83 inverted confocal microscope equipped with FV-10 software.

**Live imaging**. Fluorescent-labeled myoblast cultures were pre-differentiated for 48 h at low density (5000 cells/cm$^2$) onto Matrigel-coated plates and re-plated in Nunc Lab-Tek Chamber Slide system at a high density (75,000 cells/cm$^2$) and cultured for ~2 more days. H2B-GFP primary myoblasts were stained with SiR-Actin (Spirochrome) at 100 nM. Cells were recorded for the last 40 h of differentiation using a Nikon Ti2 microscope equipped with a motorized stage and a Yokogawa CSU-W1 spinning disk head coupled with a Prime 95 sCMOS camera (Photometrics). Cells were kept at 37 °C and 5% CO$_2$ using an incubator chamber (Okolab), while motorized stage and multipositioning images were controlled via the MetaMorph software (Molecular Devices). Specifically, for each condition and replica, four fields at ×20 magnification were recorded every 10 min.

For brightfield live imaging movies, primary myoblast cultures were seeded at 10,000 cells/cm$^2$ induced to differentiate with TGFβ1 or ITD-1. Cells were recorded for 12 h during early differentiation and for 24 h during late differentiation using Nikon Ti microscope equipped with a XY-motorized stage. Cells were kept at 37 °C and 5% CO$_2$ using an incubator chamber (Okolab), while motorized stage and multipositioning images were controlled via the MetaMorph software (Molecular Devices). Specifically, for each condition and replica four fields at ×10 magnification were recorded every 15 min. For cell speed and covered distance quantification, myoblast position was manually determined for each frame using the MetaMorph software (Molecular Devices) and then calculated using the Microsoft Excel® add-in SkyPad[58]. For cell–cell contact frequency and fusion events, myoblasts were manually followed each frame using ImageJ. For all parameters, 40 myoblasts were randomly selected per replica, for a total of 240 cells per condition.

**Western blot**. Cells were lysed with RIPA buffer (Sigma-Aldrich) supplemented with Protease and Phosphatase Inhibitor (Thermo Fisher Scientific). If needed, cytoplasmic and nuclear proteins were separated using the NE-PER Nuclear and Cytoplasmic Extraction Reagents (Thermo Fisher Scientific). Protein quantification was performed using BCA Protein Assay Kit (Thermo Fisher Scientific) and later was denatured and reduced incubating the samples with 2× Laemmli sample buffer (Santa Cruz BioTechnology) for 30 min at room temperature. Protein extracts for TGFβ1, 2, and 3 western blots were not denatured nor reduced. Equal amounts of proteins were loaded in SDS-PAGE (sodium dodecyl sulfate-polyacrylamide gel electrophoresis gel) NuPAGE™ 4–12% Bis-Tris Protein Gels (Thermo Fisher Scientific) along with molecular weight marker. Ten to thirty micrograms of total protein were loaded and transferred onto nitrocellulose membranes (Bio-Rad Laboratories). After blocking in 5% milk or BSA and 0.1% Tween-20/TBS, membranes were incubated with primary antibodies (Supplementary Table 2) overnight

and then with horseradish peroxidase-conjugated or StarBright™ (Bio-Rad Laboratories) secondary antibodies for 1 h. Specific signals were detected with a chemiluminescence system (ChemiDoc MP, Bio-Rad Laboratories).

**Bromodeoxyuridine (BrdU) assay**. Cell cultures were grown at 20,000 cells/cm$^2$ density in the growth medium on collagen-coated plates. Cells were treated with TGFβ isoforms for 1 day and then incubated with BrdU for 40 min before fixing them with 4% PFA for 5 min at RT. After a brief wash in PBS, cells were denatured with 2 M HCl for 30 min at 37 °C. To neutralize the acid, six consecutive washes in PBS of 5 min each are performed. Cells were blocked with 2% goat serum and 0.2% Tween-20 PBS for 30 min at 37 °C. Cells were incubated with primary antibody (Supplementary Table 2) for 2 h at RT and then washed three times in PBS. Secondary antibody (anti-rat IgG conjugated with Alexa Flour 546 from Thermo Fisher Scientific) was added and incubated for 45 min at RT. Before microscopic observation, three washes with PBS and Hoechst staining were performed.

**Terminal deoxynucleotidyl transferase dUTP nick-end labeling assay**. Cell cultures were grown at 20,000 cells/cm$^2$ density on collagen-coated plates and treated with TGFβ proteins for 1 day in both proliferating and differentiating conditions. Cells were fixed in 4% PFA for 20 min, washed three times with PBS, and permeabilized with 0.1% Triton X-100 and 0.1% sodium citrate PBS for 2 min on ice. After three washes in PBS, cultures were treated according to the protocol of In Situ Cell Death Detection Kit (Roche). Before observation, three washes with PBS and Hoechst staining were performed.

**Scratch-wound assay**. Primary myoblasts were plated at 30,000 cells/cm$^2$ density on collagen-coated plates. When cells reached 80% confluence, we scratched the monolayer of cells in a straight line, washed with PBS a few times, and incubated the cells in the growth medium or differentiation medium with TGFβ1. After 24 h, myoblasts were fixed with 4% PFA, stained the nuclei, and quantified the number of cells in the scar.

**Cell counting**. Quantification of cells was performed manually in a blinded manner using the point toll counter in ImageJ. The fusion index was calculated as the fraction of nuclei contained within MyHC+ myotubes, which had two or more nuclei, as compared to the number of total nuclei within each ×20 image. The differentiation index was calculated as the fraction of nuclei contained within all MyHC+ cells, including both mononuclear and multinuclear cells, as compared with the number of total nuclei within each ×20 image. Cells were considered positive for MyHC staining when the fluorescence intensity was clearly above background levels.

**Statistical analysis**. A minimum of three biological replicates was performed for the presented experiments. Error bars are standard errors. Statistical significance was assessed by the Student's $t$ test, using Microsoft Excel® and GraphPad Prism 6®. Differences were considered statistically significant at the $P < 0.05$ level. For each sample, four images were taken with a ×4, ×10, or ×20 magnification depending on the experimental design. Cell and western blot quantification and analysis were performed using ImageJ. Phalloidin staining (F-actin) images of single cells were analyzed for filament coherency using the OrientationJ plugin for ImageJ[59].

**Reporting summary**. Further information on research design is available in the Nature Research Reporting Summary linked to this article.

## Data availability

Transcriptome data have been deposited in the NCBI Gene Expression Omnibus database (https://www.ncbi.nlm.nih.gov/geo/) and is accessible through the following accession number GSE123425. Source data are provided with this paper.

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

## Acknowledgements

We thank the CyPS Facility for technical support. We thank C. Coirault, A. Brack, F. Relaix, and R. Mounier for commenting on the draft manuscript, O. Randlett for editing the revised manuscript, and C. Brun for helping with the figures. Work in the F.L.G. lab was supported by grants from the Agence Nationale pour la Recherche (ANR-14-CE11-0026 and ANR-12-JSV2-0003), and from the Association Française contre les Myopathies/AFM Telethon. F.G. was supported by a Ph.D. fellowship from the Fondation pour la Recherche Médicale (ECO20160736081). P.M.G. acknowledge the following sources for funding this study: Natural Science & Engineering Research Council (NSERC; RGPIN 435724), NSERC Canada Research Chair's Program, and the Canada First Research Excellence Fund "Medicine by Design."

## Author contributions

F.L.G. conceived and designed the study. F.G. performed most of the experiments. A.T. and C.P. performed western blotting experiments. L.G. performed qRT-PCR experiments. F.L.G performed cell culture experiments. B.C. conceived the live imaging

experiments. A.D., M.E., and P.M.G. designed, performed, and interpreted human muscle cell culture experiments. D.G.G., and D.P.M. designed, performed, and interpreted the fibroblast experiment. F.G. and F.L.G. wrote the manuscript. All authors reviewed the manuscript.

## Competing interests

The authors declare no competing interests.

## Additional information

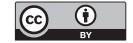

