## [Peer Review File · Nature Communications]

Reviewers' Comments:

Reviewer #1:

Remarks to the Author:

This study examines the role of the transforming growth factor-beta (TGF β) signaling in myoblast fusion stage of development and regeneration of skeletal muscles. The study analyzes the expression of TGF β during myogenesis in in vitro model and in muscle regeneration in mice, shows that application of recombinant TGF β and inhibitors of TGF β signaling suppresses and promotes myotube formation in different experimental systems and suggests that in vivo TGF β acts through regulation of actin cytoskeleton to limit myoblast fusion. Earlier reports of other groups have suggested the involvement of TGF β in muscle inflammatory response and documented inhibition of muscle regeneration and expression of various muscle-specific proteins by TGF β . While the authors of this work state that "the impact of TGF β in syncytia formation has never been investigated", TGF β has been reported to inhibit myoblast fusion quite a while ago (Olson et al, Regulation of Myogenic Differentiation by Type b Transforming Growth Factor 1986, JCB, 103, 1799-1805, not cited in the paper). Girardi et al. further develop the analysis of the effects of TGF β signaling on myoblast fusion using modern techniques and approaches. In my opinion, the most interesting finding of this paper is formation of the branched myotubes after ITD-1 application. However, this finding needs to be further developed by quantification and analysis of the underlying mechanisms. The questions addressed by this manuscript are interesting to the field, however, the conceptual advance of this study is incremental.

Specific comments.

1) Considering the existing literature showing that TGF β inhibits muscle regeneration and formation of multinucleated myotubes, the novel conclusion that "TGF β signaling limits muscle cell fusion independently of myogenic differentiation" is of critical importance for the paper. The conclusion that TGF β influences fusion rather than pre-fusion stages of myogenesis is based on uncoupling differentiation from fusion using the replating approach. The limitation of this approach is that after replating the cells at high density fusion takes two more days. Fusion event itself is known to be much faster. So, how the cells prepare for fusion during these 2 days? Can there be post-myogenin differentiation stages dependent on cell density and, perhaps, cytokine concentrations? Do the authors include pre-fusion migration, alignment, adhesion of the cells into 'fusion'? Note that TGF β has been reported to inhibit E-cadherin-based cell-cell adhesion (Vogelmann et al., J Cell Sci. 2005 Oct 15;118(Pt 20):4901-12) and, dependent on cell type and conditions, inhibit or promote cell migration (for instance, Tian, J Am Soc Nephrol. 2003 Mar;14(3):631-40). Also, in the very end of the Results, the authors state that TGF β acts upstream of "the final step of fusion" that, from the context, seems to be an actual event of myomaker- and myomerger-dependent fusion. The striking change in the cell morphology after TGF β application (Fig. 7d) also seems to suggest major changes upstream of fusion. Thus, the key conclusion seems to depend on the unclear definition of 'fusion vs. differentiation vs. the final step of fusion.

2) The qRT-PCR analysis of the time courses of expression of Tgfb1,2,3 in primary muscle cells suggests that while TGF β 3 is expressed at higher levels and is upregulated in myogenic differentiation, TGF β 1, 2 are downregulated. Despite these different expression profiles, recombinant TGF β 1,2, 3 have similar effects on myoblast fusion. This intriguing finding is not discussed. What are the concentrations of different TGF β in the conditioned medium? Are the concentrations of the recombinant TGF β used (20 ng/ml) comparable with biologically relevant concentrations?

3) I have several questions on the data in Fig. 7b that presents heatmaps of TGF β target genes, myogenic and fusion genes. (i) Sure, TGF β application boosts TGF β target genes. However, the levels of expression of TGF β target genes for the cells treated with ITD-1 do not appear significantly lower than those for the control cells (ITD-1 vs. CTR), as I would expect, if normal differentiation/fusion (in the absence of exogenous TGF β) is accompanied by TGF β signaling. (ii) It

appears that TGFb boosts expression of satellite cell marker Pax7 and suppresses expression of MyH3 and myogenin. Does not this suggest that TGFb inhibit myogenic differentiation? This is especially important because earlier studies did suggest that TGFb suppresses expression of myogenin (for instance, Zhu et al., 2004, Circulation Res. 94:617-625) (iii) Discussing this figure, the authors state that "the relative expression levels of myogenic transcription factors ... were not significantly changed". I do not see this in the figure.

4) Fig. 2d indicates that in all conditions practically all replated cells express MyHC ("~100% MyHC+"). However, the image for TGFb2 seems to suggest that many cells do not express MyHC. Why?

5) Fig. 4. According to the figure legend and the cartoon (Fig. 4A) the cells were treated with TGFb while plated at the low density but not when coplated at high density and so it appears that TGFb and ITD-1 were applied only during differentiation and not during fusion stage. Do the authors suggest that TGFb application and ITD-1 application have long term effects? This would differ from finding of Olsen et al. 1986, where transfer of TGFb-treated cultures to fresh TGFb-free media results in fusion with kinetics like those in untreated cultures. Also, the effects of TGFb can depend on the cell density (Zhu et al., 2004, Circulation Res. 94:617-625).

6) In images presented in Fig. 4b (CTR+CTR), even at 46h post replating (94 h in DM) the number and the size of myotubes are relatively low. Should not we expect (based on Fig. 2F) at this time point to see ~15% of all nuclei in 16+myotubes?

7) Fig. S1 In D inhibition is the percentage of MyHC cells is only 10% but in the images inhibition seems to be much more profound? Scale bar is in B, not in C. In D, inhibition in the percentage of MyHC cells is only 10% but in the images it seems to be much stronger. Scale bar is in B, not in C. Where is ND? What *** stands for?

8) Fig. S2c indicates that recombinant TGFb does not alter motility of myoblasts in the growth medium. Since TGFb influences myoblast fusion when applied to the replated cells in the differentiation medium, it seems more relevant to examine the TGFb effects on cell motility in the differentiation medium. The legend to this figure mentions that myoblasts were treated with TGFb1,2 or 3 but the figure presents data only for TGFb1. Why?

Minor comments.

1) Fig. 1, scale bar in D not in E.

2) The legend to Fig. 7i states that "ITD-1 treated myotubes are characterized by less nuclei per myotube" Since the Y-axis is labeled as 'Nuclei in 2-nucleated Myotubes (%)', should it be something like "ITD-1 treated myotubes are characterized by lower number of nuclei in the smallest myotubes"?

I have not found any discussion of Fig. 7 h, i and j?

3) Fig S3 The legend states that "Data are presented as mean ..." but this figure presents no quantitative data.

4) In different parts of the paper and even in the same paragraph ("Lastly, we asked ...") the same protein is referred to as 'myomergers' and as 'myomixer'.

5) Typos: 'myocytes', in a phrase "and cocultured (50/50) at high density (75000 cells/cm²) and cultured for" 'and cultured' can be deleted. 'Mysosin'.

6) there is a problem with reference 43 that misses the names of the authors and the journal Randrianarison et al., Srf controls satellite cell fusion through the maintenance of actin architecture J. Cell Biol. 217, 685–700

Reviewer #2:

Remarks to the Author:

Overall, Girardi and colleagues add to a long history of literature showing that TGF-beta inhibits myogenesis, *in vitro* and *in vivo*. The authors show that TGF-beta treatment inhibits myogenesis in mouse through the Tgfr2 and the downstream phosphorylation of Smad2/3. Injection of recombinant TGFbeta protein *in vivo* prevents efficient muscle regeneration after injury. While the notion that TGF-beta is acting at the level of fusion through modulation of actin is new, the data fall short of proving this point. The authors have the tools to build on this data and provide more direct evidence through live imaging studies of fusion and additional characterization to rule out alternate pathways such as cell size and frequency of cell-cell contacts. Understanding the functions and timing of TGF-beta during muscle regeneration would be a resource to the community and has therapeutic relevance for regenerative medicine for muscle. I think that in principle the manuscript is fitting for the journal, but it requires major revision before it can be accepted.

General Critique:

The authors describe a deficiency of TGF-beta treated cells in forming myotubes and attribute this to a fusion defect. Using canonical differentiation assays, they report fewer myonuclei per myotube formed. By live imaging assays, they also quantify a lack of fusion between TGF-beta treated and untreated cells, suggesting a cell intrinsic signaling defect. These findings are indicative of a differentiation phenotype, however, it is difficult to conclude from the data presented that the effect is fusion-specific. The authors must distinguish TGF-beta's effect on fusion from its effect on migration. This is especially salient given previous findings of a role for the TGF-beta phospho-Smad cascade in regulating the Akt hypertrophy pathway. Indeed, reduced cell size and/or cell migration can hinder contacts between myocytes. It is critical that the authors delineate hypertrophy through the Akt pathway and quantify fusion, using a live imaging approach as in Supplemental Figure S3b, to determine the percentage of cells that make contact but do not fuse. There are other significant concerns and discrepant findings that should be resolved to improve the overall coherence of the manuscript.

Specific comments:

1. Myogenic cells are indeed differentiating and fusing between days 3-6 of regeneration. Therefore, it is important to evaluate the activity of TGF-beta during this period and which cell types are affected. The quantification of TGF-beta family ligands during muscle regeneration is achieved through qRT-PCR, but must also be evaluated at the protein level by Elisa. The immunostaining of p-Smad seems to be localized in the nuclei of interstitial cells, not sub-laminar cells where myogenic cells are at these time points. The activation of p-Smad specifically in myogenic cells should be better quantified.
2. There is a discrepancy between the images shown of Pan-MyHC for TGF-beta treated myocytes and the reported differentiation index. There are clearly more single cells with background levels of staining in the TGF-beta treated samples.
3. Of concern are the reduced cell numbers and cell size in the TGF-beta treated conditions, which suggest an alternative mechanism to cell fusion.
4. Figure 3B clearly shows that the ITD-1 used is insufficient to fully inhibit pSmad2/3 signaling. Inhibitors can have off target effects. Therefore, knockdowns of Smad2/3 and/or Tgfr2 or other inhibitors of Smad2/3 are needed to prove the specificity of the signaling effect seen.
5. In Figure 4, the representative images shown for fusion events between SIR-actin H2B-GFP labeled myoblasts and TdTomato myoblasts show more SIR-actin+ TdTomato+ double positive myotubes in the TGF-beta (>3) treated samples than control (1), which disagrees with the quantification.

6. It will be critical that the authors quantify the lack of fusion observed in live imaging experiments (Figure S3b), where cells make contact but do not fuse. This would constitute direct evidence that the effect observed is fusion-specific, and does not depend on hypertrophy or migration.
7. Migration assays and cell death assays (Supplemental Figure S2) need to be performed in either proliferation conditions or in a much shorter experimental time frame. The authors should quantify the frequency of cell-cell contact after TGF-beta treatment as a control to validate that this is a fusion-specific effect.
8. Regarding the effects of TGF-beta inhibitors in human muscle stem cells in Figure 5, a gain in the size of these muscle microtissues only begins after removal from drug on day 3. Is this due to a possible refractory effect of the inhibitor? Is the Akt protein synthesis pathway driving hypertrophy after the removal of the inhibitor?
9. The transcriptomic studies shown in Figure 7b are highly variable and show a trend of higher Pax7 expression and reduced Myogenin and MyH3 expression.
10. In the experiments addressing the mechanism of actin rearrangement, ITD-1 rescued cell spreading, however, coherency of the actin cytoskeleton was not restored. This would suggest that cell spreading is more critical in ITD-1's ability to rescue fusion and differentiation. Does this equate to more cell-cell contacts? Or specifically the fusion event after cell-cell contact?
11. Since it is suggested that actin and not myomaker/myomixer is downstream of TGF-beta, the authors should try to overcome the fusion defect of TGF-beta treatment by stimulating actin through known effectors such as Rac1/RhoGTPases.
12. A discussion of Figures 7 g-j is missing from the main text.
13. The fibroblast fusion experiments does not support the conclusion that TGF-beta "signaling cascade acts independently from Myomaker and Myomixer" (pg. 7). Since signaling cascades and receptors will be different between fibroblastss and myoblasts, these experiments suggest that TGF-beta is unable to physically prevents fusion through Myomaker and Myomixer.
14. The nomenclature of Myomixer/Myomerger/MINION should be consistent throughout the text.

Reviewer #3:

Remarks to the Author:

The work of LeGrand and colleagues implicates TGF-beta in actin cytoskeletal reorganization and myoblast fusion. The authors show that exogenously added TGF-beta1, beta2 or beta3 inhibits fusion. The authors rely on a method that is suggested to uncouple differentiation from fusion. However, it remains debatable to what degree it is possible to achieve this uncoupling. The authors use a small molecular inhibitor of the TGFBR2 receptor and find that this results in larger myotubes, which is suggested to be driven by activating cell fusion. An experiment with differentially labelled cells suggests that ITD-1 acts by promoting fusion at multiple levels (cell autonomously). The response to ITD-1 is quite interesting.

Overall, the suggestion that TGF-beta itself and (and correspondingly ITD-1) are not affecting differentiation is likely an overstatement. There is a significant history dating back more than 30 years documenting the effect of TGF-beta on differentiation (e.g. see Massague J et al. PNAS 1986, Olson EN et al. J Cell Bio 1986 to the present day with more than 400 papers). Furthermore, ITD-1 does boost myogenin expression and TGF-beta1 looks like it reduces myogenin expression (Fig 7B). These data support an indirect effect through differentiation. It would be interesting to

identify fusion protein as specific targets of TGF-beta/SMAD-mediated gene expression and this information could be used to more mechanistically suggest a role for TGF-beta as a mediator of fusion. TGF-beta specific antibodies might more directly address whether blocking these antibodies actually blocks fusion itself.

Why does TGF-beta3 inhibit fusion when it goes up with differentiation? How do the authors reconcile these findings (Fig 1A which shows an increase in TGFB3 and that TGF-beta3 protein inhibits fusion in Fig 2C).

In Figure 3B, how does exogenously added TGF-beta1 overcome the loss of the receptor induced by ITD-1? This suggests that there may be other receptors mediating this effect.

What happens when ITD-1 is given to uninjured muscle or later after injury. It is not surprising that there is no increase in force so soon after injury since the muscle is still recovering from injury.

Figure 1G. The pSMAD staining is not in nuclei. This is not likely actual pSMAD staining (note the difference in the cell culture nuclei where it is in the nuclei).

Answers to the Reviewers' comments

We thank the reviewers for the positive comments and constructive criticism. We have addressed the reviewer's concerns and provide a point-by-point response below.

Reviewer #1:

This study examines the role of the transforming growth factor-beta (TGF β) signaling in myoblast fusion stage of development and regeneration of skeletal muscles. The study analyzes the expression of TGF β during myogenesis in in vitro model and in muscle regeneration in mice, shows that application of recombinant TGF β and inhibitors of TGF β signaling suppresses and promotes myotube formation in different experimental systems and suggests that in vivo TGF β acts through regulation of actin cytoskeleton to limit myoblast fusion. Earlier reports of other groups have suggested the involvement of TGF β in muscle inflammatory response and documented inhibition of muscle regeneration and expression of various muscle-specific proteins by TGF β . While the authors of this work state that "the impact of TGF β in syncytia formation has never been investigated", TGF β has been reported to inhibit myoblast fusion quite a while ago (Olson et al, Regulation of Myogenic Differentiation by Type β Transforming Growth Factor 1986, JCB, 103, 1799-1805, not cited in the paper). Girardi et al. further develop the analysis of the effects of TGF β signaling on myoblast fusion using modern techniques and approaches. In my opinion, the most interesting finding of this paper is formation of the branched myotubes after ITD-1 application. However, this finding needs to be further developed by quantification and analysis of the underlying mechanisms. The questions addressed by this manuscript are interesting to the field, however, the conceptual advance of this study is incremental.

We thank the reviewer for evaluating our work. We removed the phrase highlighted by the reviewer, and we referenced additional previous work on TGF β in the introduction.

Specific comments.

1) Considering the existing literature showing that TGF β inhibits muscle regeneration and formation of multinucleated myotubes, the novel conclusion that "TGF β signaling limits muscle cell fusion independently of myogenic differentiation" is of critical importance for the paper. The conclusion that TGF β influences fusion rather than pre-fusion stages of myogenesis is based on uncoupling differentiation from fusion using the replating approach. The limitation of this approach is that after replating the cells at high density fusion takes two more days. Fusion event itself is known to be much faster. So, how the cells prepare for fusion during these 2 days?

Before fusing, replated cells first have to adhere to the substrate, and then will move toward each other to establish cell-cell contact and ultimately fuse. Of note, while the first fusion events happen in the first hours after replating, many other events occur within the two days period, for myotube to accrete nuclei and mature.

To understand how muscle cells prepare for fusion, we performed live imaging of pre-fusing myocytes. We observed that muscle cells move and contact each other extensively before fusing (Figure 4).

Can there be post-myogenin differentiation stages dependent on cell density and, perhaps, cytokine concentrations?

Fusion itself is regulated by cell plating density and inhibited by low cell proximity. It is then not feasible to evaluate the effect of lowering cell density on the activity of a negative regulator of fusion.

Do the authors include pre-fusion migration, alignment, adhesion of the cells into 'fusion'? Note that TGF β has been reported to inhibit E-cadherin-based cell-cell adhesion (Vogelmann et al., J Cell Sci. 2005 Oct 15;118(Pt 20):4901-12) and, dependent on cell type and conditions, inhibit or promote cell migration (for instance, Tian, J Am Soc Nephrol. 2003 Mar;14(3):631-40).

We indeed previously included pre-fusion cell motility within the "fusion" process. Thank to the reviewer comments we are now more careful and discuss cell motility and adhesion as preliminary phases to the fusion process. We now present data demonstrating that TGF β signaling does not prevent the frequency of cell-to-cell contacts (Fig.4) nor does it impact muscle cell migration (Fig.S3).

Also, in the very end of the Results, the authors state that TGF β acts upstream of "the final step of fusion" that, from the context, seems to be an actual event of myomaker- and myomerger-dependent fusion. The striking change in the cell morphology after TGF β application (Fig. 7d) also seems to suggest major changes upstream of fusion. Thus, the key conclusion seems to depend on the unclear definition of 'fusion vs. differentiation vs. the final step of fusion.

This is a poor choice of words from our part. By upstream, we meant "before". Our data suggest TGF β acts on fusion through impacting actin cytoskeleton, and not by modulating the "final" step of fusion. The final step is Myomaker/Myomerger-driven membrane coalescence. We corrected our statement in the text.

2) The qRT-PCR analysis of the time courses of expression of Tgfb1,2,3 in primary muscle cells suggests that while TGF β 3 is expressed at higher levels and is upregulated in myogenic differentiation, TGF β 1,2 are downregulated. Despite these different expression profiles, recombinant TGF β 1,2, 3 have similar effects on myoblast fusion. This intriguing finding is not discussed.

We observed TGF β 3 expression going up during myoblast differentiation. In our opinion, this fits with the result showing that TGF β signaling is still active in multinucleated muscle cells both in vitro (Fig. 1) and in vivo (Fig. 2). We propose that TGF β 3 may be the main source of TGF β ligand expressed by differentiated cells, to prevent unscheduled fusion between myotubes or myofibers. We improved the discussion and included this specific issue.

What are the concentrations of different TGF β in the conditioned medium? Are the concentrations of the recombinant TGF β used (20 ng/ml) comparable with biologically relevant concentrations?

The recombinant TGF β proteins are added in the medium at 20ng/ml. This concentration has been used in other culture systems. It is adequate since it allows for activation of the signaling pathway without influencing cell proliferation/death.

It is not possible to compare physiological concentrations of a particular secreted protein with something we use in vitro. How can a biological concentration of TGF β could be evaluated? Also TGF β biological activity varies due to the latent/active states of the ligand.

3) I have several questions on the data in Fig. 7b that presents heatmaps of TGF β target genes, myogenic and fusion genes. (i) Sure, TGF β application boosts TGF β target genes. However, the levels of expression of TGF β target genes for the cells treated with ITD-1 do not appear significantly lower than those for the control cells (ITD-1 vs. CTR), as I would expect, if normal differentiation/fusion (in the absence of exogenous TGF β) is accompanied by TGF β signaling. (ii) It appears that TGF β boosts expression of satellite cell marker Pax7 and suppresses expression of MyH3 and myogenin. Does not this suggest that TGF β inhibit myogenic differentiation? This is especially important because earlier studies did suggest that TGF β suppresses expression of myogenin (for instance, Zhu et al., 2004, Circulation Res. 94:617-625) (iii) Discussing this figure, the authors state that “the relative expression levels of myogenic transcription factors ... were not significantly changed”. I do not see this in the figure.

The heatmap in figure 7 (now 9) highlighted very slight variations in gene expression that are not significant. Since the expression levels of the transcripts we selected for building the heatmap were not different between conditions, the software we used created “shades” of red from the fluctuations in the triplicate values.

We performed a more controlled analysis of the changes in gene expression from the transcriptomic data and generated a heatmap using Prism 6 software. As shown in Figure 9a, the expression levels of expression of TGF β target genes change following either TGF β or ITD-1 treatments, while the expression levels of myogenic markers and membrane effectors of fusion do not.

We provide the reviewers with a version of the previous heatmap, with the values in gene expression for each replicate as Fig.Rev.1. The expression levels of *Myh3* and *Myogenin* are not changed following TGF β stimulation (see also table 3 for complete dataset).

4) Fig. 2d indicates that in all conditions practically all replated cells express MyHC (“~100% MyHC+”). However, the image for TGF β 2 seems to suggest that many cells do not express MyHC. Why?

This is due to the fact that the staining intensity for MyHC proteins varies with the state of maturation of the differentiated muscle cells. As such, mononucleated cells appears weakly positive for MF20, compared to multinucleated cells.

We provide the reviewer with a support figure Fig.Rev.2a showing that myocytes, at 72h of differentiation express significant amounts of MyHC proteins and are considered “positive” compared to the “negative” muscle cells in growth conditions or in 24hours differentiation conditions while expressing lower amounts compared to matured myotubes. Fig.Rev.2b is a single-color of Figure 3d, showing that most cells are positive for MyHC immunoreactivity.

5) Fig. 4. According to the figure legend and the cartoon (Fig. 4A) the cells were treated with TGF β while plated at the low density but not when coplated at high density and so it appears that TGF β and ITD-1 were applied only during differentiation and not during fusion stage. Do the authors suggest that TGF β application and ITD-1 application have long term effects?

Yes, our experiments show that exposition to TGF β and/or ITD-1 have long-term effects. Preliminary results obtained in our lab, that were not included in the manuscript, demonstrated that TGF β “washout” does not revert the already activated intracellular signaling pathway.

We provide the reviewers with Fig.Rev.3. that show evidence of long-term effects of TGF β application. TGF β target gene *Klf10* increase in expression is maintained in stimulated cells 3 days after medium washout.

This would differ from finding of Olsen et al. 1986, where transfer of TGFb-treated cultures to fresh TGFb-free media results in fusion with kinetics like those in untreated cultures.

This is pertinent, but the data by Olson et al, was obtained in C2C12, while we work with primary muscle cells. C2C12 differentiate and fuse with slower kinetics compared to adult primary myoblasts. A difference in the sensitivity to TGF β addition or in the persistence of the signaling is conceivable.

Also, the effects of TGFb can depend on the cell density (Zhu et al., 2004, Circulation Res. 94:617-625).

We agree with the reviewer. However, it is not possible to test these rules by analogy with other cell types since the rate of fusion depends on cell density.

6) In images presented in Fig. 4b (CTR+CTR), even at 46h post replating (94 h in DM) the number and the size of myotubes are relatively low. Should not we expect (based on Fig. 2F) at this time point to see ~15% of all nuclei in 16+myotubes?

The reviewer's concern is correct, however there are multiple explanations for this potential incoherency. First of all, in this setup only half of the nuclei are visualized (H2B-GFP), thus it is hard to have a complete picture of the myotube size and nuclei number. Second, primary cells deriving from different animals do not display the same "fusion efficiency" as compared to a single cell line, consequently, it is conceivable to obtain a slightly reduced fusion. Third, SiR-Actin might reduce cell activity being a compound that binds to actin filaments (although it does not alter or bias the experiment as we can notice from our movies). These issues, together with the fact that here we are quantifying only the heterologous fusion and not the total fusion, lead us to consider the fusion index obtained as reasonable and expected.

7) Fig. S1 In D inhibition is the percentage of MyHC cells is only 10% but in the images inhibition seems to be much more profound? Scale bar is in B, not in C. Where is ND? What *** stands for?

The inhibition appears stronger due to the lack of mature myotubes in TGF β -treated cultured (as in comment 4). We corrected the scale bar and statistics legends. In the previous version of our manuscript. "N.D." meant "not detected". We corrected the figure using "N.S." for "not significant".

8) Fig. S2c indicates that recombinant TGFb does not alter motility of myoblasts in the growth medium. Since TGFb influences myoblast fusion when applied to the replated cells in the differentiation medium, it seems more relevant to examine the TGFb effects on cell motility in the differentiation medium. The legend to this figure mentions that myoblasts were treated with TGFb1,2 or 3 but the figure presents data only for TGFb1. Why?

As suggested, we performed scratch-wound assays in differentiation medium (Figure S3c) and found no impact of TGF β 1 on cell migration in either proliferation or differentiation media.

Minor comments.

1) Fig. 1, scale bar in D not in E.

2) The legend to Fig. 7i states that "ITD-1 treated myotubes are characterized by less nuclei per myotube" Since the Y-axis is labeled as 'Nuclei in 2-nucleated Myotubes (%)', should it be

something like “ITD-1 treated myotubes are characterized by lower number of nuclei in the smallest myotubes“?. I have not found any discussion of Fig. 7 h, i and j?

3) Fig S3 The legend states that “Data are presented as mean ...” but this figure presents no quantitative data.

4) In different parts of the paper and even in the same paragraph (“Lastly, we asked ...”) the same protein is referred to as ‘myomerger’ and as ‘myomixer’.

5) Typos: ‘myocytes’, in a phrase “and cocultured (50/50) at high density (75000 cells/cm²) and cultured for” ‘and cultured’ can be deleted. ‘Mysosin’.

6) there is a problem with reference 43 that misses the names of the authors and the journal Randrianarison et al., Srf controls satellite cell fusion through the maintenance of actin architecture J. Cell Biol. 217, 685–700

We thank the reviewer for his/her very careful review. We made all the corrections in the text.

Reviewer #2

Overall, Girardi and colleagues add to a long history of literature showing that TGF-beta inhibits myogenesis, in vitro and in vivo. The authors show that TGF-beta treatment inhibits myogenesis in mouse through the Tgfr2 and the downstream phosphorylation of Smad2/3. Injection of recombinant TGFbeta protein in vivo prevents efficient muscle regeneration after injury.

While the notion that TGF-beta is acting at the level of fusion through modulation of actin is new, the data fall short of proving this point. The authors have the tools to build on this data and provide more direct evidence through live imaging studies of fusion and additional characterization to rule out alternate pathways such as cell size and frequency of cell-cell contacts. Understanding the functions and timing of TGF-beta during muscle regeneration would be a resource to the community and has therapeutic relevance for regenerative medicine for muscle. I think that in principle the manuscript is fitting for the journal, but it requires major revision before it can be accepted.

We thank the reviewer for evaluating our work and suggesting important improvements.

General Critique:

The authors describe a deficiency of TGF-beta treated cells in forming myotubes and attribute this to a fusion defect. Using canonical differentiation assays, they report fewer myonuclei per myotube formed. By live imaging assays, they also quantify a lack of fusion between TGF-beta treated and untreated cells, suggesting a cell intrinsic signaling defect. These findings are indicative of a differentiation phenotype, however, it is difficult to conclude from the data presented that the effect is fusion-specific. The authors must distinguish TGF-beta's effect on fusion from its effect on migration. This is especially salient given previous findings of a role for the TGF-beta phospho-Smad cascade in regulating the Akt hypertrophy pathway. Indeed, reduced cell size and/or cell migration can hinder contacts between myocytes. It is critical that the authors delineate hypertrophy through the Akt pathway and quantify fusion, using a live imaging approach as in Supplemental Figure S3b, to determine the percentage of cells that make contact but do not fuse. There are other significant concerns and discrepant findings that should be resolved to improve the overall coherence of the manuscript.

As suggested by the reviewer, we studied TGFβ's effect on cell migration/motility and on cell contacts (reported in the new Fig. 4). We recorded primary muscle cells in differentiation medium (containing TGFβ1 protein or not) in two different time frames (early differentiation: before cell fusion, and late differentiation: during syncytium formation) (Fig. 4b). We observed that muscle cells move and contact each other extensively before and during fusion (supplementary movies 1 to 4).

We show that TGFβ signaling does not impact cell speed/motility (Fig. 4c), and that TGFβ1 treatment does not influence the frequency of cell-to-cell contacts in early (Fig.4c) or late (Fig. 4d) time frames. Moreover, as per the reviewer suggestion, we determined the percentage of cells that make contact but do not fuse (Fig 4d) and now show that TGFβ1 stimulation reduce the proportion of contacts that lead to a fusion event. We calculated that while in control conditions, only one fourth of cell-to-cell contacts were not leading to a fusion event, TGFβ1 treatment resulted in the majority of cell-to-cell contacts to be unproductive in terms of fusion.

Specific comments:

1. Myogenic cells are indeed differentiating and fusing between days 3-6 of regeneration. Therefore, it is important to evaluate the activity of TGF-beta during this period and which cell types are affected.

To identify which myogenic cell types are affected by TGF β signaling, we performed co-immunolocalization for phospho-SMAD3 and muscle cell markers on cryosections from uninjured skeletal muscle tissue and from regenerating muscle tissue at two relevant time points (4d.p.i. when cells are actively fusing, and 7d.p.i. when the fusion process is almost completed).

As shown in the new Fig. 2, we found TGF β signaling to be active in quiescent and activated satellite cells (PAX7-expressing), to be down-regulated in MYOGENIN⁺ differentiating cells and to be active in the myonuclei of DYSTROPHIN⁺ newly-formed myofibers. This new data strengthens our initial observation that TGF β signaling is active in differentiated fused muscle cells *in vitro* (Fig. 1d) and highlights the myogenic cells affected by TGF β signaling *in vivo*.

The quantification of TGF-beta family ligands during muscle regeneration is achieved through qRT-PCR, but must also be evaluated at the protein level by Elisa.

We were unable to have *in vivo* Elisa to work in our hands (regenerating samples produce a high background). On the other hand, to answer the reviewer question, we evaluated TGF β family ligands protein expression by western blotting. We present the results in the new Fig. S1. The protein expression profiles match the RNA expression profiles.

The immunostaining of p-Smad seems to be localized in the nuclei of interstitial cells, not sub-laminar cells where myogenic cells are at these time points. The activation of p-Smad specifically in myogenic cells should be better quantified.

Our previous immunofluorescences were sub-optimal, including high non-specific staining. We improved our immunostaining technique for phospho-SMAD3 and present new images showing the activation of TGF β signaling specifically in myogenic cells at the cellular level on resting and regenerating muscle tissue cryosections (Fig. 2).

2. There is a discrepancy between the images shown of Pan-MyHC for TGF-beta treated myocytes and the reported differentiation index. There are clearly more single cells with background levels of staining in the TGF-beta treated samples.

This is a question similar to Reviewer #1 fifth comment. As previously discussed, the vast majority of mononucleated cells express pan-MyHC, but with an intensity level lower than mature multinucleated cells.

We provide the reviewer with a support figure Fig.Rev.2a showing that myocytes, at 72h of differentiation express significant amounts of MyHC proteins and are considered "positive" compared to the "negative" muscle cells in growth conditions or in 24hours differentiation conditions while expressing lower amounts compared to matured myotubes. Fig.Rev.2b is a single-color of Figure 3d, showing that most cells are positive for MyHC immunoreactivity.

3. Of concern are the reduced cell numbers and cell size in the TGF-beta treated conditions, which suggest an alternative mechanism to cell fusion.

The reduced cell size is a consequence of reduced fusion. Primary muscle cells have low cytoplasm/nuclei ratio, that hugely increase following the formation of multinucleated myotubes. As such each fused myotube bears more cell volume compared to unfused myocytes.

The apparent reduced density instead is a consequence of a bias introduced by the myotubes. Myotubes in fact gather together and cluster more nuclei compared to mononucleated cells in the same area. However, there are no differences in the overall densities between control and

treatment as we plate the same number of cells and we specifically demonstrated that neither proliferation nor cell death are altered by TGF β activation.

4. Figure 3B clearly shows that the ITD-1 used is insufficient to fully inhibit pSmad2/3 signaling. Inhibitors can have off target effects. Therefore, knockdowns of Smad2/3 and/or Tgfr2 or other inhibitors of Smad2/3 are needed to prove the specificity of the signaling effect seen.

While we never stated nor did we tried to fully inhibit SMAD signaling with ITD1, we tested another TGF β pathway inhibitor as suggested by the reviewer. As shown in figure S4, treatment of re-plated myocytes with the drug SB-431542, a potent inhibitor of TGFBR1, increased muscle cell fusion.

5. In Figure 4, the representative images shown for fusion events between SIR-actin H2B-GFP labeled myoblasts and TdTomato myoblasts show more SIR-actin+ TdTomato+ double positive myotubes in the TGF-beta (>3) treated samples than control (1), which disagrees with the quantification.

We now present a more representative image in the new Fig. 6. Moreover, we provide for the reviewer the Fig. Rev.4 where we specifically show that this issue was simply related to SiR-Actin signal intensity. Indeed, in Fig.Rev.4, we can notice that the number of heterologous myotubes (myotube co-expressing SiR-Actin and tdTOMATO) are reduced in the TGF β 1+CTR co-cultures, compared to CTR+CTR.

6. It will be critical that the authors quantify the lack of fusion observed in live imaging experiments (Figure S3b), where cells make contact but do not fuse. This would constitute direct evidence that the effect observed is fusion-specific, and does not depend on hypertrophy or migration.

To answer the reviewer comment, we performed extensive analysis of new live imaging experiments. As shown in the new figure 4, we quantified muscle cell motility in control and TGF β -treated cultures. We observed that TGF β signaling does not impact cell movement or speed, nor did it prevent cells to make contact (Fig. 4d). However, we could observe that TGF β stimulation resulted in a strong increase in the proportion of “unproductive” cell contacts (a.k.a. that does not lead to a fusion event) (Fig. 4d). These data add a new level of comprehension to our story.

7. Migration assays and cell death assays (Supplemental Figure S2) need to be performed in either proliferation conditions or in a much shorter experimental time frame.

As suggested, we performed migration assays and cell death assays in both proliferation and differentiation conditions. The data is presented in new Fig. S3.

The authors should quantify the frequency of cell-cell contact after TGF-beta treatment as a control to validate that this is a fusion-specific effect.

As suggested, we quantified the frequency of cell-cell contact after TGF β treatment and found no differences with control cells (Fig. 4c)

8. Regarding the effects of TGF-beta inhibitors in human muscle stem cells in Figure 5, a gain in the size of these muscle microtissues only begins after removal from drug on day 3. Is this due to a possible refractory effect of the inhibitor? Is the Akt protein synthesis pathway driving hypertrophy after the removal of the inhibitor?

To answer the reviewer's question, we tested if TGF β signaling impact AKT/mTOR pathway in myocytes. As shown in new Fig. S5, western-blot experiments for p-AKT and p-mTOR show no activation/inhibition of the pathway by either TGF β or ITD-1.

9. The transcriptomic studies shown in Figure 7b are highly variable and show a trend of higher Pax7 expression and reduced Myogenin and MyH3 expression.

This is a question similar to Reviewer #1 third comment. The heatmap in figure 7 (now 9) highlighted very little changes in gene expression that are not significant. Since was due to a poor handling of the heatmap algorithm.

We performed a more controlled analysis of the changes in gene expression from the transcriptomic data and generated a heatmap using Prism 6 software. As shown in Figure 9a, the expression levels of expression of TGF β target genes change following either TGF β or ITD-1 treatments, while the expression levels of myogenic markers and membrane effectors of fusion do not. We provide the reviewers with a version of the previous heatmap, with the value in gene expression for each replicate as Fig.Rev.1. The expression levels of myogenic markers *Pax7*, *Myh3* and *Myogenin* are not changed following TGF β stimulation.

10. In the experiments addressing the mechanism of actin rearrangement, ITD-1 rescued cell spreading, however, coherency of the actin cytoskeleton was not restored. This would suggest that cell spreading is more critical in ITD-1's ability to rescue fusion and differentiation. Does this equate to more cell-cell contacts? Or specifically the fusion event after cell-cell contact?

To answer the reviewer question, we quantified the frequency of cell-cell contact in myocytes cultures treated with ITD-1. As shown in Fig.9f, we found no differences compared to control cells. This data suggests that ITD-1 specifically acts on the fusion event.

11. Since it is suggested that actin and not myomaker/myomixer is downstream of TGF-beta, the authors should try to overcome the fusion defect of TGF-beta treatment by stimulating actin through known effectors such as Rac1/RhoGTPases.

We performed the suggested experiments but did not obtain any interesting results. The "RhoA Activator" from Cytoskeleton Inc, did not have any potency on muscle cells. Experimental activation of F-Actin polymerization by the drug Jaspakinolide resulted in a high mortality *in vitro*, precluding evaluation of the fusion process.

12. A discussion of Figures 7 g-j is missing from the main text.

We apologize for this absence, previously due to a cut/paste error. Discussion of results has been added back to the text.

13. The fibroblast fusion experiments does not support the conclusion that TGF-beta "signaling cascade acts independently from Myomaker and Myomixer" (pg. 7). Since signaling cascades and receptors will be different between fibroblastss and myoblasts, these experiments suggest that TGF-beta is unable to physically prevents fusion through Myomaker and Myomixer.

We thank the reviewer for the clarification. We changed the text using the reviewer's suggestion.

14. The nomenclature of Myomixer/Myomerger/MINION should be consistent throughout the text.

We verified consistency of the nomenclature.

Reviewer #3

The work of LeGrand and colleagues implicates TGF-beta in actin cytoskeletal reorganization and myoblast fusion. The authors show that exogenously added TGF-beta1, beta2 or beta3 inhibits fusion. The authors rely on a method that is suggested to uncouple differentiation from fusion. However, it remains debatable to what degree it is possible to achieve this uncoupling. The authors use a small molecular inhibitor of the TGFBR2 receptor and find that this results in larger myotubes, which is suggested to be driven by activating cell fusion. An experiment with differentially labelled cells suggests that ITD-1 acts by promoting fusion at multiple levels (cell autonomously). The response to ITD-1 is quite interesting.

We thank the reviewer for evaluating our manuscript.

Overall, the suggestion that TGF-beta itself and (and correspondingly ITD-1) are not affecting differentiation is likely an overstatement. There is a significant history dating back more than 30 years documenting the effect of TGF-beta on differentiation (e.g. see Massague J et al. PNAS 1986, Olson EN et al. J Cell Bio 1986 to the present day with more than 400 papers). Furthermore, ITD-1 does boost myogenin expression and TGF-beta1 looks like it reduces myogenin expression (Fig 7B). These data support an indirect effect through differentiation.

We apologize for the likely overstatement. This was not our intention. We now include the early work on TGF β in the introduction. We also improved our wording and propose that while inhibition of TGF β is a necessary step for muscle cell differentiation, this signaling also plays a role in the fusion process.

We now demonstrate that TGF β signaling does not impact Myogenin expression in differentiated cells.

1. As also noted by the other reviewers, the heatmap in Fig.7b was misleading and highlighted very little changes in gene expression that are not significant. Since was due to a poor handling of the heatmap algorithm. We now present a heatmap in fig. 9a showing that TGF β and ITD-1 treatment do not impact the expression of myogenic genes among them Myogenin (as well as the transcriptomic dataset associated with the manuscript, Table 3). We provide the reviewers with a version of the previous heatmap, with the value in gene expression for each replicate as Fig.Rev.1. The expression levels of myogenic markers *Pax7*, *Myh3* and *Myogenin* are not changed following TGF β stimulation.

2. We performed new experiments to ascertain that TGF β signaling do not act through Myogenin expression to limit muscle cell fusion. As shown in figure 3c, pre-differentiated myocytes replated at high density were stimulated with TGF β 1 protein. We analyzed gene expression after 18 hours and observed that while the TGF β target gene *Smad7* expression was strongly activated by the treatment (demonstrating activation of the pathway) there was no significant differences in Myogenin expression compared to control. These new data confirms that TGF β has a specific role on cell fusion in differentiated myocytes..

It would be interesting to identify fusion protein as specific targets of TGF-beta/SMAD-mediated gene expression and this information could be used to more mechanistically suggest a role for TGF-beta as a mediator of fusion.

As suggested by the reviewer, we investigated the TGF β target genes (Table 3) to identify proteins with roles in the fusion process. We selected transcripts that exhibited strong changes in gene expression in our transcriptomic analysis. We found the TGF β target genes *Timp1* and *Arhgef6* have an effect on muscle cell fusion, without impacting cell differentiation. The results

are presented in new Fig. S7. Together with a better evaluation of cell motility, we demonstrate a role for TGF β in regulating the fusion process.

TGF-beta specific antibodies might more directly address whether blocking these antibodies actually blocks fusion itself.

We performed the suggested experiment. Unfortunately, we did not observe any differences between control and treated cells. We include Fig.Rev.5 for the reviewer. We can hypothesize that TGF β signaling is already very high in proliferation cells, and that the signal persists in differentiating cell in vitro even if the ligand availability is perturbed.

Why does TGF-beta3 inhibit fusion when it goes up with differentiation? How do the authors reconcile these findings (Fig 1A which shows an increase in TGFB3 and that TGF-beta3 protein inhibits fusion in Fig 2C).

We observed TGFb3 expression going up during myoblast differentiation. In our opinion, this fits with the result showing that TGF β signaling is still active in multinucleated muscle cells both in vitro (Fig. 1) and in vivo (Fig. 2). We propose that TGFb3 may be the main source of TGFb ligand expressed by differentiated cells, to prevent unscheduled fusion between myotubes or myofibers. We improved the discussion and included this specific issue.

In Figure 3B, how does exogenously added TGF-beta1 overcome the loss of the receptor induced by ITD-1? This suggests that there may be other receptors mediating this effect.

We tested another TGF β pathway inhibitor as suggested by the reviewer. As shown in figure S4, treatment of re-plated myocytes with the drug SB-431542, a potent inhibitor of TGFBR1, increased muscle cell fusion.

What happens when ITD-1 is given to uninjured muscle or later after injury. It is not surprising that there is no increase in force so soon after injury since the muscle is still recovering from injury.

While we found the reviewer comments very interesting, we did not investigate the effects of ITD-1 on uninjured muscles. We believe this is out of the scope of the manuscript. Such experiments would investigate the role of TGF β signaling in skeletal muscle resting state, while our work focus on tissue regeneration and cell fusion.

Figure 1G. The pSMAD staining is not in nuclei. This is not likely actual pSMAD staining (note the difference in the cell culture nuclei where it is in the nuclei).

We agree on the fact that our previous immunofluorescences were sub-optimal, including high non-specific stainings. We improved our immunostaining technique for phospho-SMAD3 and present new images showing the activation of TGF β signaling specifically in myogenic cells in the new Fig. 2.

Figures for the reviewers.

Figure Rev 1 | Expression values onto the previous microarray heatmap.

a, The heatmap presented in our first submitted draft created a concern for the reviewers (Figure 7b). In fact, although we stated that no significant differences were detected for the *Myogenin* transcript levels upon TGFβ stimulation or inhibition, the heatmap misled the reviewers due to unproper color representation of the heatmap highlighting changes in gene expression of 1%. In support of our word, we represent the heatmap with the related values (Raw data available on NCBI Gene Expression Omnibus database; see Materials and Methods for accession number). Moreover, as you can notice from figure 9, a new and more representative heatmap has been integrated in the paper.

Figure Rev 2 | Myotube maturation and Pan-MyHC (MF-20) staining.

a, Primary myoblasts were induced to differentiate for 3 days and stopped at 4 different time points; 0, 24, 48 and 72 hours of differentiation. Cells were stained with Pan-MyHC to visualize differentiated myocytes/myotubes. From these immunofluorescences we can notice the difference between MyHC- cells (#) and MyHC+ cells (*). Although MyHC+ cells exhibit different staining intensities, they must be considered differentiated cells. In fact, different intensities correlate with different maturation stages, as exemplified in the figures 48h and 72h.

b, Immunofluorescent staining for the Myosin Heavy Chain isoforms (Pan-MyHC) of re-plated primary myocytes cultured for 48 hours (as in Figure 2d). In accordance with what described above, the Pan-MyHC staining reveals that practically all cells are positive (~100%). This concept can be extended also to figure Supp 1. Scale bars: **a**, 400 μ m. **b**, 200 μ m.

Figure Rev 3 | TGFβ long-term effect.

In reviewer #1, question (5) we have been asked if we propose that TGFβ has a long-term effect. The answer is yes; we show here that a single pulse of TGFβ (or ITD-1) prior to fusion is enough and sufficient to impact the fusion process (Figure 4). **a**, As support, here we show a qRT-PCR analysis for *Klf10* (TGFβ target gene) performed on myoblast differentiated for 3 days and treated with a single pulse of TGFβ1 at the onset of differentiation (Medium was refreshed every 24 hours). **b**, qRT-PCR analysis reveals that TGFβ signaling pathway is still active 3 days after the stimulation.

Figure Rev 4 | Heterologous myotube quantification.

a, As indicated by the reviewer #2, the live imaging frame figures for Control (CTR+CTR) and TGFβ1 (CTR+TGFβ1) were not clear. This is mainly due to the low intensity of the SiR-Actin signal. We thus increased the red intensity obtaining a better representative figure. **b**, In fact, if we specifically quantify the heterologous myotubes (*) from these pictures we can notice 15 heterologous myotubes (*) in Control and 7 in TGFβ1, numbers that better represent the quantification shown in figure 4. Scale bars: **a**, 200μm. **b**, 200μm.

Figure Rev 5 | TGF β isoform neutralizing antibody.

a, As suggested by reviewers #1, we tried to neutralize the TGF β isoform protein present in the medium or secreted from the cells. Primary myoblasts were induced to differentiate in medium containing TGF β 1, 2 and 3 neutralizing antibody (R&D System, MAB1835-SP) at a final concentration of 1 μ g/ml. **b**, Immunofluorescent staining for Pan-MyHC of 3-days differentiated myotubes. In our hand, no evident phenotypical differences were observed after 6 replicas, indicating that the neutralization of TGF β protein present in the medium is not enough to increase fusion.

Reviewers' Comments:

Reviewer #1:

Remarks to the Author:

The new data and editing have considerably strengthened the paper, addressed my concerns and clarified the take-home message. This is an important contribution to the field and I have only minor suggestions/comments.

1) I am not sure what the experiments with Latrunculin B (Fig. 9g-h) add to the paper. Suppression of myoblast (C2C12) fusion by Latrunculin B has been reported before (Nowak et al., 2009 JCS 122, 3282) and this inhibition was found to be accompanied by a loss of cell mobility. Under conditions used here Latrunculin B has completely blocked formation of large myotubes for both untreated and ITD-1 treated myoblasts. What is the evidence that Latrunculin B inhibits the same fusion or pre-fusion stage that is promoted by ITD-1? Also, in Fig. 9h what N.D. stands for? Should the sentence "... if the fusogenic effect of ITD-1 can be rescued by treating cells with Latrunculin ..." be edited to "... if the fusogenic effect of ITD-1 can be suppressed by treating cells with Latrunculin ...".

2) In Fig. 6b in a and b panels the authors count hours in different ways with t= 6h in b corresponding to t= 54h in a.

3) In Fig. 6e the numbers of myotube-myotube fusion events after the treatments are normalized to that in the control. How often are these events observed in the control?

4) The sentence "Taken together our data indicate that TGF β signaling depends on the actin cytoskeleton to exerts its function on cell fusion." is confusing and needs editing.

Reviewer #2:

Remarks to the Author:

Please see attachment:

Girardi et al Nat Comm Revision

Reviewer #4:

Remarks to the Author:

In my opinion, the authors responded to the concerns raised by the reviewer#3.

They faithfully cite early work to show that TGF β inhibits terminal differentiation.

In Fig.3E, data showing uncoupling differentiation and fusion of primary muscle cells are presented, and myogenin expression was conserved even after TGF β treatment.

They studied downstream molecules of TGF β signaling such as actin. Furthermore, although effects are rather weak, the authors demonstrate potential downstream molecules of TGF β in fusion, such as Timp1 and Arhgef6.

Two independent chemical TGF β inhibitors are used. Imaging data are also helpful to see visually. Branching phenotype by TGF β inhibition is also interesting.

Answers to the Reviewers' comments

Reviewer #1 (Remarks to the Author):

The new data and editing have considerably strengthened the paper, addressed my concerns and clarified the take-home message. This is an important contribution to the field and I have only minor suggestions/comments.

We are thankful to the reviewer for his/her evaluation of our work.

1) I am not sure what the experiments with Latrunculin B (Fig. 9g-h) add to the paper. Suppression of myoblast (C2C12) fusion by Latrunculin B has been reported before (Nowak et al., 2009 JCS 122, 3282) and this inhibition was found to be accompanied by a loss of cell mobility.

These experiments were suggested to us by the colleagues who read the manuscript before submission. The goal was to assess if the hyper fusion phenotype induced by ITD1 is absent when the actin cytoskeleton is compromised. To this aim, we dosed the concentration of Latrunculin B that allows most of the cells to fuse (fusion index is 74% in control cultures and 67% in Latrunculin-treated cells) (fig. 9h) but prevent the formation of large myotubes (fig. 9j). Thus, our data demonstrate that ITD-1 cannot promote fusion in cells that are capable of fusion but have perturbed actin dynamics. We added this last sentence to the manuscript on page 10.

Under conditions used here Latrunculin B has completely blocked formation of large myotubes for both untreated and ITD-1 treated myoblasts. What is the evidence that Latrunculin B inhibits the same fusion or pre-fusion stage that is promoted by ITD-1?

In this experimental setup, 95% of the myocytes are terminally differentiated. The dosage of Latrunculin B allows most cells to establish contacts and fuse. This indicate that, in our hands, Latrunculin B treatment does not act on the first steps of myogenic differentiation/ fusion but prevents the formation of large myotubes which is the phenotype associated with inhibition of TGFbeta signaling.

Also, in Fig. 9h what N.D. stands for? Should the sentence "... if the fusogenic effect of ITD-1 can be rescued by treating cells with Latrunculin ..." be edited to "... if the fusogenic effect of ITD-1 can be suppressed by treating cells with Latrunculin ...".

We changed "N.D." into "N.S." for "Not significant". We edited the phrase as suggested.

2) In Fig. 6b in a and b panels the authors count hours in different ways with t= 6h in b corresponding to t= 54h in a.

We updated figure 6 so the hours match between panels 6a and 6b.

3) In Fig. 6e the numbers of myotube-myotube fusion events after the treatments are normalized to that in the control. How often are these events observed in the control?

In control conditions, the myotube-myotube fusion events were observed at the frequency of 0,25 events per hour (out of 4 20x fields).

4) The sentence "Taken together our data indicate that TGFb signaling depends on the actin cytoskeleton to exerts its function on cell fusion." is confusing and needs editing.

We edited the sentence into "Taken together our data indicate that TGFβ signaling limits cell fusion by preventing actin cytoskeleton remodeling".

Reviewer #4 (Remarks to the Author):

In my opinion, the authors responded to the concerns raised by the reviewer#3. They faithfully cite early work to show that TGFbeta inhibits terminal differentiation. In Fig.3E, data showing uncoupling differentiation and fusion of primary muscle cells are presented, and myogenin expression was conserved even after TGFbeta treatment. They studied downstream molecules of TGFbeta signaling such as actin. Furthermore, although effects are rather weak, the authors demonstrate potential downstream molecules of TGFbeta in fusion, such as Timp1 and Arhgef6. Two independent chemical TGFbeta inhibitors are used. Imaging data are also helpful to see visually. Branching phenotype by TGFbeta inhibition is also interesting.

We thank the reviewer for his/her careful review.

Reviewer #2 (Remarks to the Author):

In this manuscript, Girardi and colleagues the authors characterize the effect of TGFbeta signaling on inhibition of skeletal muscle cell fusion. The work has been improved by the performance of additional experiments and the results support the conclusions. However, there is an issue of novelty. Most of the findings have previously been shown by others in the field.

- TGFbeta signaling was shown to have a negative effect on skeletal muscle cell differentiation and fusion (Massegue et al. 1986 & Olson et al., 1986).
- Signaling through TGFB receptors and downstream Smad proteins has also been shown to repress function of specific myogenic factors involved specifically in differentiation such as MyoD (Liu et al, 2001).
- Additionally, in vivo TGFbeta treatment was shown to decrease regenerative capacity after muscle injury (Li et al., 2004),
- TGFbeta pathway inhibition has been extensively shown to increase fiber size (Egerman et al., 2013).

We agree with the reviewer that the role of TGFbeta in blocking muscle cell differentiation has been well established by previous work. We state this in the introduction of our manuscript.

However, the previous work only suggested that TGFbeta signaling might regulate fusion. There is no direct evidence that fusion is perturbed in the previous data. There was not enough knowledge about fusion then to fully determine the role of TGFbeta signaling. Previously reported "fusion" phenotype was actually related to the inhibition of the upstream process of myogenic differentiation.

Regarding the issue of novelty and from a more conceptual perspective for the field, we have some of the first data for a mechanistic brake on the fusion process.

We also demonstrate that TGFbeta signaling does not prevent differentiated muscle cells to establish contacts but render these contacts infructuous in fusion. Furthermore, and as stated by the other reviewers, our description of myotube-to-myotube fusion events is novel and important for the field.

Additionally, a potentially novel finding that is underdeveloped is the effect of TGFbeta on actin dynamics and organization, and the opposite effect on muscle cells compared to other cell types such as cancer cells and epithelial cells, which increase invasiveness and invadopodia upon TGFbeta treatment (Edlund et al, 2002; Bolan et al., 1996). Indeed, it seems likely that it is the lack of such protrusions that impedes fusion. What is lacking, however, is an analysis of the effectors and pathways leading to this phenomenon and how it differs with other cell types. This novel finding and a mechanistic analysis would greatly increase the impact of this manuscript.

We agree that investigating the effectors and pathways controlling the formation of Actin-rich protrusions is interesting and of key interest for the field. However, this represent a many-years study

and not a question we can answer within the span of a revision. As such the suggested mechanistic analysis is out of the scope of the study.

Within the revised manuscript, we already provide potential downstream molecules of TGFbeta in fusion (Timp1 and Arhgef6, both linked to Actin dynamics) and pave the way for future work that will span the coming years in our lab and others.

Other potential novel points of this study are the purported dissociation of the effect of TGFbeta on fusion from differentiation, two processes that are hard to separate. Notably, past literature shows a clear effect of TGFbeta on differentiation, eg., through direct repression of MyoD. It is unclear that the authors have excluded this possibility and they should modify their claims accordingly.

We certainly do not pretend that the previous work is wrong. TGFbeta clearly acts on the differentiation step, as shown by many previous studies. This is why we perform most of our experiments on differentiated muscle cells. We show that (i) in "replated" terminally-differentiated myocytes, TGFbeta stimulation does not influence Myogenin expression by RT-qPCR (Fig.3) and that (ii) TGFbeta stimulation of differentiated myocytes does not influence the expression of MyoD, Myogenin and of their target genes Mymk and Myomerger/Myomiwer/MINION by transcriptomic assay (Fig.9).

Note that the English is very poor and the manuscript needs extensive editing.

This comment is difficult to comprehend since two native English writers contributed to writing the manuscript and reviewed the text (Dr. Gilbert and Dr. Millay). We asked Dr. Owen Randlett (native English writer and PI at the Institut NeuroMyoGène) to read the manuscript and correct grammatical errors. The changes suggested by Dr. Randlett are in green font throughout the text. Moreover, the Nature Communications team include editors, specialized in English language, who will correct the remaining grammatical errors.

Although the manuscript is improved overall, some issues remain:

1. Due to prior literature in the field, specifically the effect of TGFbeta on MyoD repression, an effect on differentiation needs to be excluded. This could be achieved by analyzing in vitro live cell imaging data in the new figure 4, which is convincing in showing inhibition of cell fusion, but lacks an analysis of differentiation markers.

We performed the suggested experiments. As per the reviewer suggestion, we differentiated cells cultured in the same condition as for the live imaging experiment and ;

1. Stained 48hours-differentiated cells with anti-MyoD1 antibodies (see Figure 1 for the Reviewer).
2. Extracted RNA from 60hours-differentiated cells and performed qRT-PCR for Myogenin expression.

The new data is presented in figure 4b and demonstrate that TGFbeta does not change the proportion of MyoD+ cells and does not change the level of Myogenin gene expression 60hrs following a single TGFB1 pulse. We suggest in the text that the inhibitory effect of TGFbeta on muscle cell differentiation (that we document in figure S3) is partially compensated in our culture conditions, at least for the differentiation markers analyzed.

Additionally, it should be noted that most of the in vitro experiments were not performed using the method in Figure 3, which dissociates fusion from differentiation. This should be stated in the text.

We chose to not use method of figure 3 to do figure 4 as the replating at high density impedes a clear tracking of cells motion and contacts. This was a fortunate idea, since it allowed us to document the absence of differences between control and TGFB-treated cells during early differentiation. We added the justification in the text.

2. A major issue remains that in vitro experiments are not well connected to the in vivo phenotype.

We actually believe our in vitro experiments are very well connected to the in vivo phenotype. First, in both cases, persistence of p-SMAD3 immunoreactivity can be observed in differentiated multinucleated muscle cells (Fig. 1 and Fig. 2). The data suggest that TGFbeta signaling is still active in fusing cells. This, we propose, to prevent excessive fusion. Second, modulation of TGFbeta signaling both in vivo and in vitro resulted in alteration in the fusion process, in a similar fashion.

While the authors show staining for p-SMAD3 in regenerating muscle sections, the cell types affected by TGF-beta/SMAD signaling during regeneration remain poorly characterized. In fact, in the new figure 2 b-d, the majority of centrally nucleated muscle fibers show positive p-SMAD3 staining in their nuclei, which suggests possible additional roles of SMAD3 in myofibers. Could this prevent additional cells from fusing into these still immature myofibers?

The finding that centrally nucleated myofibers have active TGF-beta signaling is, in our opinion, one of the important pieces of data within our manuscript. As we wrote in the discussion, we think that a basal level of TGF-beta signaling activity in differentiated muscle cells is required to prevent excessive fusion. As suggested by the reviewer, we added the phrase “The persistence of p-SMAD3 signaling in newly-formed (centrally-nucleated) myofibers during muscle tissue regeneration suggest TGFβ signaling prevent additional cells from fusing into these still immature myofibers.” to the discussion.

Furthermore, p-SMAD3 staining surrounding Myogenin+ cells at day 4 suggests that these cells may not be sensitive to local TGFbeta1.

As previously noted by the reviewer, downregulation of TGFbeta signaling is required for myoblast to differentiate. This is in accordance with MYOGENIN+ cells being p-SMAD3-negative in vivo.

The authors should perform p-SMAD3 staining on their TGF-beta1 and ITD-1 treated muscles to confirm that Myogenin+ cells are indeed sensitive to TGF-beta1 treatment in vivo and that ITD-1 is able to reduce p-SMAD3 in myoblasts.

As suggested by the reviewer, we performed p-SMAD3 staining on our TGFbeta1 and ITD-1 treated muscles. Unfortunately, it was not possible to get clear nuclear immunostaining on TGFbeta1-treated slides. We spent a long time trying to, but the fibrosis and inflammation that accompany TGFβ1 injection prevented immunolocalization of Myogenin.

See Figure 2 for the Reviewer

Then, while we were successful in visualizing p-SMAD3 along with Pax7 on ITD-1-treated samples, we did not find a significant difference in the number of p-SMAD+ myoblasts compared to control. This was understandable, since the cells that were subjected to ITD-1 action at 3d.p.i., when the compound was injected, did fuse to form the new myofibers and participated to the increased number of myonuclei in ITD-1-treated samples.

3. Quantification of the pan-MyHC experiments is still problematic. The use of Boolean counting is clearly masking a phenotype of reduced expression. This is used to argue that fusion is the major deficit after TGF-beta1 treatment. The data in Fig. 3d-g, S2b-e should include a distribution of the average staining intensity of each myotube.

We performed the proposed quantification of pan-MyHC staining intensity in single myotubes. The data is presented in new Figure S3.

4. For experiments reporting cell counting, the authors should make a statement as to how cells were considered “positive” and whether these quantifications were performed blinded.

We added a section describing the counting method.

Minor comments

Fig. 7 is a major experiment that shows human relevance but lacks detailed description in the main text and requires discussion of methodology and significance.

We added a detailed description of the hMMT method, and a phrase related to the significance at the end of the discussion.

Fig. S8 is too zoomed out to appreciate differences in fusion. A zoomed in inset would be a good addition.

Since figure S8 contains high definition images and will be available as a pdf, readers will be able to easily zoom in.

Grammatical errors are present throughout the text.

We asked Owen Randlett (native English speaker and PI at the Institut NeuroMyoGène) to read the manuscript and correct grammatical errors. The changes suggested by Dr. Randlett are in green font throughout the text.

Figures for the reviewer.

Figure 1

Primary myoblasts were induced to differentiate for 48hours. Cells were fixed and stained for MYOD1 proteins (red) and Sarcomeric Alpha Actin (SAA, green).

Figure 2

Adult murine *tibialis anterior* (TA) muscles were subjected to CTX injury and regenerating tissues were injected intramuscularly with either TGF β 1 proteins or ITD-1 compound 3 days after damage. Regenerated muscle tissues were sampled 7 days after injury. Cryosections were stained with antibodies against DYSTROPHIN (green) PAX7 or MYOGENIN (red) and p-SMAD3 (white). While fluorescent stainings were successful on ITD1-treated muscles, the injection of TGF β 1 prevented any immunolocalizations on tissue sections.

Reviewers' Comments:

Reviewer #1:

Remarks to the Author:

The revision and the reply have fully addressed my questions. The findings are novel and convincing, and deepen understanding of the mechanisms by which TGF β signaling regulates myoblast fusion.

Reviewer #2:

Remarks to the Author:

The revised manuscript by Girardi et al. has largely addressed my concerns and improved the interpretation and discussion of the results. The authors now convincingly separate the effects of TGF- β signaling on myogenic differentiation and actin-reorganization during fusion in vitro. While it is difficult to tease these apart in vivo, the phenotypes after TGF- β or inhibitor injection are consistent with the interpretation. I appreciate the extra efforts by the authors to address my comments. I also appreciate the addition of text regarding the sequential expression of ligands by different cell types being required to prevent premature fusion and also act on myotubes to prevent fusion between syncytia. This is a novel concept for skeletal muscle regeneration. The revised manuscript is acceptable for publication.

Reviewer #1 (Remarks to the Author):

The revision and the reply have fully addressed my questions. The findings are novel and convincing, and deepen understanding of the mechanisms by which TGF β signaling regulates myoblast fusion.

Reviewer #2 (Remarks to the Author):

The revised manuscript by Girardi et al. has largely addressed my concerns and improved the interpretation and discussion of the results. The authors now convincingly separate the effects of TGF- β signaling on myogenic differentiation and actin-reorganization during fusion in vitro. While it is difficult to tease these apart in vivo, the phenotypes after TGF- β or inhibitor injection are consistent with the interpretation. I appreciate the extra efforts by the authors to address my comments. I also appreciate the addition of text regarding the sequential expression of ligands by different cell types being required to prevent premature fusion and also act on myotubes to prevent fusion between syncytia. This is a novel concept for skeletal muscle regeneration. The revised manuscript is acceptable for publication.

We thank the reviewers for their evaluation of our work